Review article

# Insights into silicon cycling from ice sheet to coastal ocean from isotope geochemistry
Katharine R. Hendry [1,2] ✉, Felipe Sales de Freitas[3], Sandra Arndt[3,4], Alexander Beaton[5], Lisa Friberg[2], Jade E. Hatton[2,6,7], Jonathan R. Hawkings [4,8], Rhiannon L. Jones[1], Jeffrey W. Krause [9,10], Lorenz Meire [11,12], Hong Chin Ng[2], Helena Pryer[13], Sarah Tingey[4], Sebastiaan J. van de Velde [14,15,16], Jemma Wadham[4], Tong Wang[2] & E. Malcolm S. Woodward [17]

The polar regions are biologically productive and play a critical role in regional and global biogeochemical cycling. A key nutrient is dissolved silicon, required for the growth of siliceous phytoplankton, diatoms, which form an important component of polar ecosystems. Glacial weathering is thought to be an important dissolved silicon source to coastal waters, especially critical in regions experiencing seasonal silicon limitation of diatom growth. However, complex physical and biogeochemical interactions in fjords and coastal regions modulate the downstream supply of dissolved and particulate nutrients, including silicon. Here, we review the biogeochemical complexities of glaciated margins and the insights into this system that silicon isotope geochemistry offer. We show that stable and radioisotopic measurements and biogeochemical numerical modelling provide a quantitative mechanistic understanding of subglacial silica mobilisation and its cycling across the land-ocean continuum. Subglacial weathering produces isotopically light amorphous silica, which dissolves in seawater to release dissolved silicon. Our findings show that isotopically light, detrital silica, likely containing glacial material, reaches the ocean and there could support a substantial proportion of diatom productivity, especially in the Arctic. Outstanding questions about silicon cycling in these crucial environments will be addressed through novel and cross-discipline approaches that overcome traditionally viewed ecosystem boundaries.

Diatoms form the base of polar marine ecosystems, often despite low and biologically-limiting near-surface concentrations of the critical nutrient dissolved silicon or silicic acid (DSi) relative to biological requirements, especially in the Arctic and subarctic[1–3]. Subglacial weathering supplies both DSi and reactive amorphous silica (ASi), which releases DSi on dissolution[4]. However, the supply of glacial silica to coastal margins remains a contentious subject due to outstanding questions surrounding the complex transformations involved with the take-up and release of DSi as it transits from the subglacial environment through glacial streams and fjords into the coastal ocean[5,6]. Here, we review the recent advances that silicon isotope geochemistry and biogeochemical modelling have offered into complex Si cycling processes in glaciated ocean margins. Stable Si isotopes ($\delta^{30}$Si) provide a mechanistic understanding of subglacial ASi formation and dissolution in pelagic and benthic systems

across the land-ocean continuum[7–11]. We have begun to quantify DSi utilisation by diatoms, and the role of inorganic particulate material in supplying DSi to silicon-limited fjord and coastal populations, by coupling stable isotopes and radioactive $^{32}$Si experiments[12,13]. By combining DSi and stable Si isotope measurements with reaction-transport modelling, we have been able to quantitatively constrain the dominant known (or hypothesised) processes acting upon these complex environments. This includes the use of reaction-transport models for understanding water-particle interactions, reaction pathways/rates, and seafloor exchange fluxes during the early diagenesis of Si in coastal sediments[14–16]. As such, cross-discipline research and use of new generation models and other novel technologies, including nutrient and other biogeochemical (BGC) sensors, will be critical for understanding the cycling of Si, and other elements, in these vulnerable critical zones.

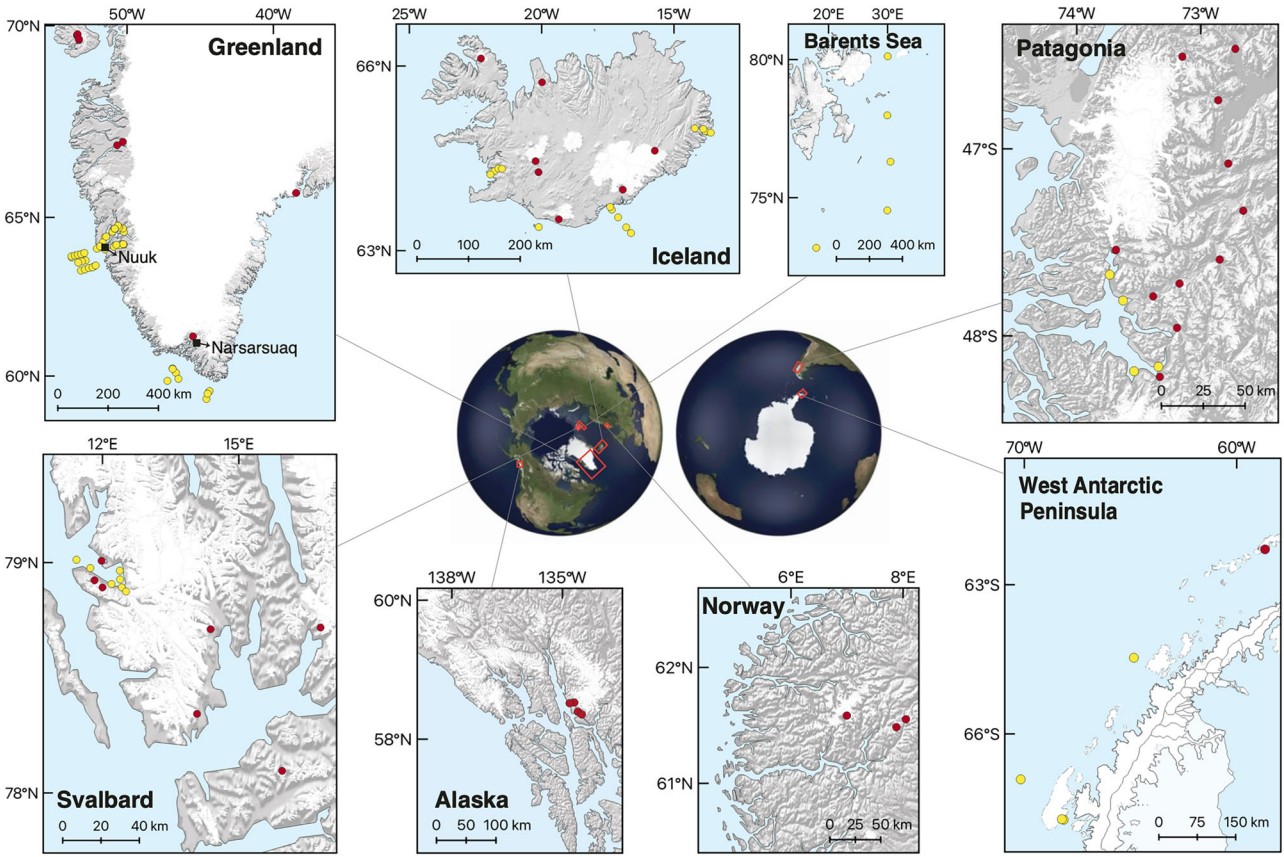

**Fig. 1 | Map locations of the study sites referenced in this review.** Red symbols show terrestrial locations (i.e., freshwater); yellow symbols show fjord and marine locations. Data from[4,8,9,12,13,15,22,46–49,53]. Maps produced in QGIS using Natural Earth vector datasets and the Randolf Glaciers Inventory (RGI Consortium, 2023[73]).

## Generation of dissolved and particulate silica during subglacial weathering processes

Subglacial weathering under glaciers and ice sheets is known to release biologically important inorganic and organic nutrients and reactive particulate material that have the potential to influence downstream biological production and elemental inventories[5,17–19]. Glacial meltwaters contains DSi at concentrations that can be significantly higher than background surface seawater concentrations in polar regions, especially in the Arctic, subarctic and biologically productive subantarctic islands[20]. In addition to DSi, subglacial silicate weathering beneath glaciers and ice sheets produces a form of reactive amorphous silica (ASi) that is dissolvable in seawater and can represent up to 95% of the potentially bioavailable silicon (i.e., DSi) exported from glacial river systems[4,8,10,21]. ASi is operationally defined as silica dissolved using an alkaline digestion, either through heating the sample with either sodium carbonate (0.1 M $Na_2CO_3$) or sodium hydroxide (0.2 M NaOH), where the former generally produces similar or slightly higher concentrations compared to the latter method[4,7,10]. A compilation of ASi extraction data from glaciated regions (see map in Fig. 1) reveals that ASi concentrations in glacial sediments range by two orders of magnitude from 3 to ~800 μmol g$^{-1}$ and illustrates geographical variability that indicates a lithological or hydrological control (Fig. 2 and Supplementary Fig. S1).

Simple incubation experiments, comparing the dissolution rate of glacial sediments from a Greenlandic catchment with and without alkaline digestion, reveal that this ASi dissolves readily in low nutrient ([DSi] <1 μM) seawater at 18 °C. Approximately 20% of the available ASi dissolved after two weeks releasing DSi, and it was hypothesised that within approximately one year nearly all ASi would dissolve, if dissolution rates remained on a

similar trajectory[4]. More recent experiments have confirmed this rate of ASi dissolution from Greenlandic glacier-derived sediments in ambient fjord waters[21,22]. They indicate that this dissolution appears to be a function of sediment loading but is independent of light-dark cycles and temperature[22]. These incubations were carried out on frozen material, and we recommend that further experiments using fresh material would help better understand the role of biological processes and potential light and temperature dependence of ASi dissolution. Regardless, investigations to date support the hypothesis that the dissolution of ASi sourced from meltwaters is an important source of DSi to downstream ecosystems, supporting siliceous organisms including diatoms[12,20].

Stable silicon isotope measurements (Box 1) are a useful tool in identifying both the source of the subglacially-derived ASi and the mechanism behind its formation. Pan-Arctic glacial rivers have significantly lower isotopic values than non-glacial rivers (average $\delta^{30}Si_{DSi}$ = +0.16‰ and +1.38‰ for glacial and non-glacial rivers respectively[8]), and these low isotopic values has also been found in subglacial meltwaters from polythermal (wet-based) Antarctic systems[8–10,23,24]. Where data are available, the isotopic signal of DSi in glacial runoff appears to change over a melt season, trending towards lower $\delta^{30}Si_{DSi}$ values later in the season, reflecting the evolution of hydrological conditions and channel connectivity in the subglacial environment[10]. The isotopic composition of ASi, $\delta^{30}Si_{ASi}$, is lower than the surrounding sediments and bedrock, but—in contrast to the dissolved phase—remains consistent throughout a melt season[7,10]. Glacial river $\delta^{30}Si_{ASi}$ shows a weak relationship with the concentration of suspended particulate matter, but does not show a clear link with lithology[8].

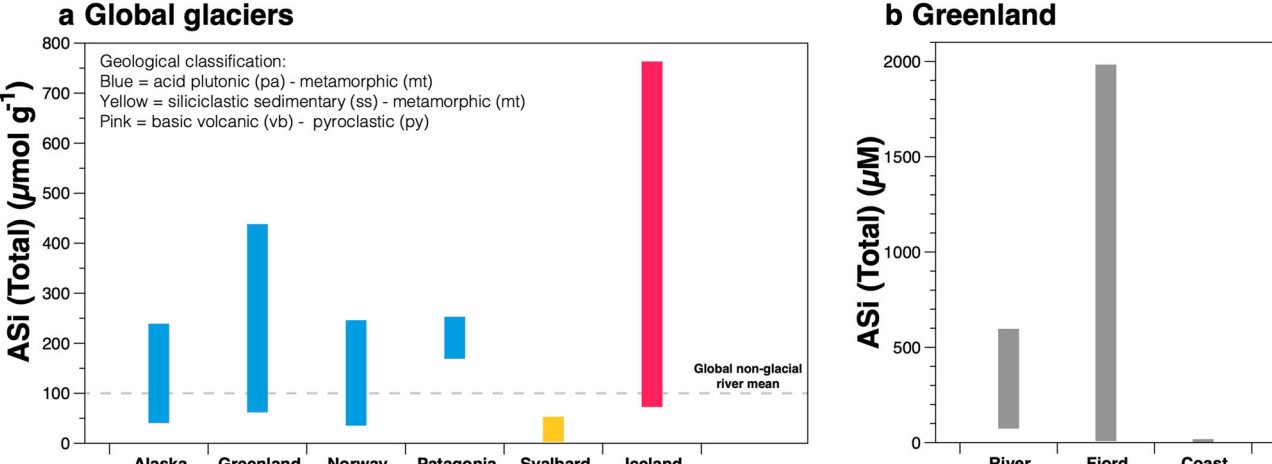

**Fig. 2 | Amorphous silica extracted from glacial sediments.** Full range of amorphous silica (ASi) extracted from suspended sediments filtered from (**a**) glacially-fed rivers and (**b**) glacially-fed rivers, glaciated fjord and coastal waters in West Greenland. Data from[4,8,13,15,22,29,53]. These plots show the total ASi extracted using a weak-alkaline leach, but additional information can be obtained about other reactive silica pools using sequential extractions (See Supplementary Notes and Fig. S1). Note: for Patagonia glacial bulk meltwater samples, only > 0.45 μm data are plotted (see discussion on colloidal material in the main text).

The light isotopic signature that characterises glacial silica, both in dissolved and amorphous phases, not only provides a "chemical fingerprint" of different meltwater types but also points towards the nature of subglacial weathering processes. The isotopically light compositions may arise from physical weathering processes that create highly reactive, disturbed mineral surfaces[25], coupled with chemical precipitation-hydrolysis reactions at grain boundaries[26,27], which result in the formation of isotopically light amorphous crusts[11,24]. Rock crushing experiments support the involvement of mechano-chemical processes in ASi formation in subglacial environments[11]. Pre-weathered glacial till, collected from a Greenlandic catchment and formed predominantly from unsorted quartz and plagioclase, was mechanically milled for different periods of time before undergoing dissolution in ultrapure fresh water; the milled material was then dried, re-crushed and reacted with ultrapure water again to mimic ongoing subglacial crushing processes. These experiments demonstrated that the DSi initially released into the ultrapure water is very isotopically light (as low as −2.12‰) and that the light isotopic composition of DSi could be recreated by re-crushing and creating freshly disturbed surface layers[11]. Whilst alkaline digestions indicated ASi was present in all (crushed and uncrushed) samples, there were significantly higher concentrations in samples that had undergone more intense (longer) grinding. The mean grain size was smaller in the samples that were crushed for longer, but did not seem to result in greater fractionation, attributed to isotopic heterogeneity within—or pre-weathering of—the parent rock, and/or formation and dissolution of ASi. ASi formed in the crushing process has a consistent isotopic signature (−0.22 ± 0.15‰ 2 SD)[11].

The relationship between glacial weathering and silica isotopic composition, however, appears to be more complex in some glaciated settings likely related to geology. In Patagonia, the concentration and isotopic composition of DSi in glacier-fed river systems has been shown to be highly dependent on the filtration methodology in addition to glacial cover: the <0.45 μm fraction was characterised by isotopically light Si that was linearly related to the percentage glacial cover in the watershed primarily due to a previously unidentified colloidal-nanoparticulate (0.02–0.45 μm) phase[28,29]. This colloidal material was only present in glacially-sourced rivers and is thought to be comprised predominantly of feldspars. The potential bioavailability of silica and trace metals from this glaciogenic colloidal-nanoparticulate phase in downstream ecosystems remains unknown.

## Box 1 | The use of silicon isotope geochemistry in polar science

- Silicon comprises of three stable isotopes ($^{28}$Si, $^{29}$Si and $^{30}$Si). The stable silicon isotopic composition of natural waters (meltwaters, fjord waters, and seawater) reveals information about the processes that release or take up dissolved silicon (DSi).
- Stable isotopic compositions are measured relative to a reference standard (usually NBS28, also known as RM8546) and are denoted by standard delta notation, $\delta^{30}$Si (dissolved fraction $\delta^{30}$Si$_{DSi}$ and amorphous fraction $\delta^{30}$Si$_{ASi}$)[7].
- Biological uptake of DSi to form biogenic silica (BSi) or opal preferentially takes up the lighter Si isotopes. Diatoms fractionate silicon isotopes with a fractionation factor, $^{30}\varepsilon$, of approximately −1.1 ‰[77]. Siliceous sponges have a much wider range of apparent fractionation factors that are a function of the concentration of available DSi[78].
- Abiotic processes can also fractionate stable Si isotopes. Benthic uptake of DSi into authigenic clays and adsorption onto iron and manganese oxyhydroxides tends to result in strong negative isotopic fractionation[43].
- The $^{32}$Si radioisotope of Si can be used to assign rates to removal and release processes. For example, the rate of BSi production by diatoms can be assessed through additions of D$^{32}$Si to seawater samples that are left to incubate at suitable temperatures and light levels before filtration; after equilibration with the daughter product, $^{32}$P, the particulate $^{32}$Si activity is measured using beta counting[79] or liquid scintillation counting[32].
- Benthic reaction-transport models are useful tools to help fully disentangle Si isotopic dynamics at the sediment-water interface, for example in understanding Si-Fe interactions at redox boundaries[14,15,46].

### Silicon cycling along the land-ocean continuum

We now have a stronger grasp of the production and supply mechanisms behind the flow of dissolved Si and particulate silica from the subglacial environment. Although there are inherent geochemical links between

subglacial silicate weathering, oxidative weathering and atmospheric $CO_2$ uptake/release[30], we also need to consider downstream processes for a full picture of how glacially mobilised Si links with carbon cycling and climate. Below we assess quantitatively and mechanistically how this glacially-sourced DSi and ASi impacts downstream ecosystems and biological carbon drawdown.

## Fjords

In most glaciated environments, the DSi and ASi released in glacial runoff will flow into the coastal ocean via fjords, which are typically separated from the continental shelf by a shallower sill. Herein lies a key issue: fjords are not only conduits but also active and highly heterogeneous biogeochemical reactors, with multiple input sources, including non-glacial rivers and the ocean itself e.g., see ref. 31, and abiotic and biotic processes occurring across strong but temporally and spatially variable salinity gradients that form a continuum with the open ocean[13]. Also, in the case of land-terminating glacial fjords, any glacially-sourced nutrients will also be exposed to additional processing in the pro-glacial watershed system, which could include glacial lakes in addition to rivers[31]. A proportion of both DSi and ASi will likely be trapped, at least in part, within the fjord (and/or pro-glacial) environment before reaching—and so potentially supporting—coastal ecosystems. Silicic acid, DSi (including that from the dissolution of ASi), will be taken up by fjord-dwelling diatoms to form their biogenic silica (BSi) shells or frustules. However, it will also adsorb onto reactive mineral phases that are also generated by subglacial weathering and/or formed in glaciogenic marine sediments (e.g., iron and manganese oxyhydroxides[15]). Particulate amorphous silica (e.g., ASi), which can dissolve to release DSi in seawater, will sink and can potentially be buried in the seafloor of the fjord. The residence time of particulate material in fjords and its spatial reach downstream are clearly critical parameters to constrain in order to determine the extent to which the dissolution of detrital material releases biologically important nutrients into seawater[21,22]. These processes are also superimposed on strong geochemical gradients between the glacial freshwater and marine endmembers, and dynamic physical (buoyancy, wind, iceberg-driven) mixing.

Within complex glacially-fed fjord systems, there are some important questions to address: what is the relative importance of glacially exported Si and upwelled Si from a marine benthic endmember as Si sources to diatoms? Specifically, how important is ASi dissolution in supplying DSi to diatoms within fjords? Why does DSi appear to behave conservatively in fjords? Is BSi dissolution reduced in these environments due to low temperatures impacting bacterial activity and physicochemical processes, thus increasing the importance of ASi sources of DSi? How does Si cycling in glacially-fed fjord systems differ from non-glaciated estuaries and coastal regions? Answering these questions requires both a concerted effort at increasing observations and field-based experimental studies, but also incorporating biogeochemical modelling.

Whether marine or glacial sources are important for supporting fjord phytoplankton communities appears to have a strong dependence on the surrounding conditions of the fjord: the greater the concentration and flux of DSi from glacial sources relative to marine sources, and its supply relative to the supply of other nutrients (i.e., the elemental stoichiometry), the greater its relative importance will be. A greater supply of Si, from either glaciers or marine upwelling, will only impact phytoplankton growth in regions where there is silicon limitation or co-limitation. Hence, for glacial systems that are feeding fjord ecosystems with low—potentially limiting—ambient DSi concentrations, where low-DSi Atlantic waters dominate marine inputs, the Si-rich meltwaters are likely to be an important supply mechanism, such as found in southwest Greenland[4,13,20,32,33]. This input could increase in importance in the future, given the declining DSi over time reported in the Atlantic-influenced sectors of the subarctic and Arctic[34,35]. However, further to the northwest of Greenland and into the Canadian Arctic Archipelago (CAA), where Pacific-sourced marine waters supply a higher concentration of DSi into fjords, the relative importance of glacial silica is likely to be lower[36,37].

The role of ASi dissolution in fjord Si cycling is less straightforward and remains unquantified. As discussed above, laboratory experiments have shown that ASi is soluble in seawater, and it would be expected that, as the salinity increases across the fjord environmental gradient, such dissolution would increase[4]. There is some field evidence for an increase in fjord DSi concentrations with salinity (where salinity <10) (e.g., ref. 38) rather than decrease, which could be explained by ASi dissolution. However, these apparent reverse trends could also be a result of complex heterogeneities in natural systems and sampling limitations[5]. It appears highly unlikely that DSi behaves conservatively in these fjordic environments: diatoms dominate the phytoplankton assemblages in many of these localities (e.g., ref. 39), and several abiotic exchange reactions (adsorption, desorption, precipitation, and dissolution) are likely to be active in the vicinity of the reactive mineral surfaces of fresh, glacially-derived particles. Assessment of reactive silica cycling in these environments is particularly challenging given that the alkaline extraction method used will not differentiate between glacially-derived ASi and diatom-derived BSi. Furthermore, while apparent linear relationships between DSi and salinity within fjords could at face value indicate largely conservative behaviour, the highly scattered nature of these relationships highlight the operation of critical non-conservative reactions, especially near source inputs[13].

Stable Si isotopes can provide novel insight into these challenging questions. If the glacial and marine endmembers are known, within ranges at least, then a plot of fjord water column samples in 'isotope space' should reveal a linear relationship between $\delta^{30}Si_{DSi}$ and the inverse of DSi concentration (i.e., 1/[DSi]) if Si cycling within the fjord was driven by simple conservative mixing. However, this appears not to be the case, at least for two well-studied fjord systems in southwest Greenland (Fig. 3a, b)[13]. We can take an additional step and assume that a considerable amount of the DSi uptake is due to diatom production. A simple isotope fractionation model shows that the $\delta^{30}Si_{DSi}$ of water column samples within these fjords cannot be explained by diatom uptake of marine-sourced DSi alone[13]. Furthermore, these models suggest that the $\delta^{30}Si_{DSi}$ observations cannot be explained simply by a mixture of marine and glacially-derived DSi, but also require additional enrichment of light isotopic material, consistent with the dissolution of ASi and/or other detrital material within the fjord, or the addition of another—yet unquantified—endmember.

## Fjord and continental margin sediments

The benthic environment forms another important, and poorly understood, component of the land-ocean continuum Si cycle. The dissolution and precipitation reactions that release and remove DSi from within the water column, respectively, and will continue at the sediment-water interface and during early diagenesis in shallow sediments[14,40]. Once silica settles on the seafloor, its residence time dramatically increases, increasing the influence of diagenetic reaction processes on nutrient cycling. Whilst silicate weathering and authigenic clay formation (termed reverse weathering) have classically been studied in low latitude delta settings (e.g., refs. 41–43), there is increasing evidence of dynamic Si cycling during early diagenesis in mid-latitude[44,45] and high latitude marine sediments[14,46,47]. The imbalance between dissolution and precipitation reactions leads to a build-up of DSi within pore fluids, with concentrations classically increasing to an asymptotic maximum. The pore fluid DSi can then flux out of the sediments into the overlying bottom waters via molecular diffusion, given the concentration gradient, and via other mechanisms, such as bioturbation, bioirrigation and wave-induced pore fluid flushing[14].

A trend is emerging with the increasing availability of pore fluid and sediment data from glaciated regions: the build-up of DSi in pore fluids increases with distance away from a glacial terminus, manifesting in higher asymptotic DSi concentrations (and higher benthic DSi fluxes) with sediment depth in distal versus proximal sites (Fig. 4a–d). Furthermore, the isotopic composition of the DSi in pore fluids in proximal sites is substantially heavier than distal locations and fjords fed by non-glacial rivers studied in Greenland and southern Patagonia[15,48] and high latitude coastal and open ocean

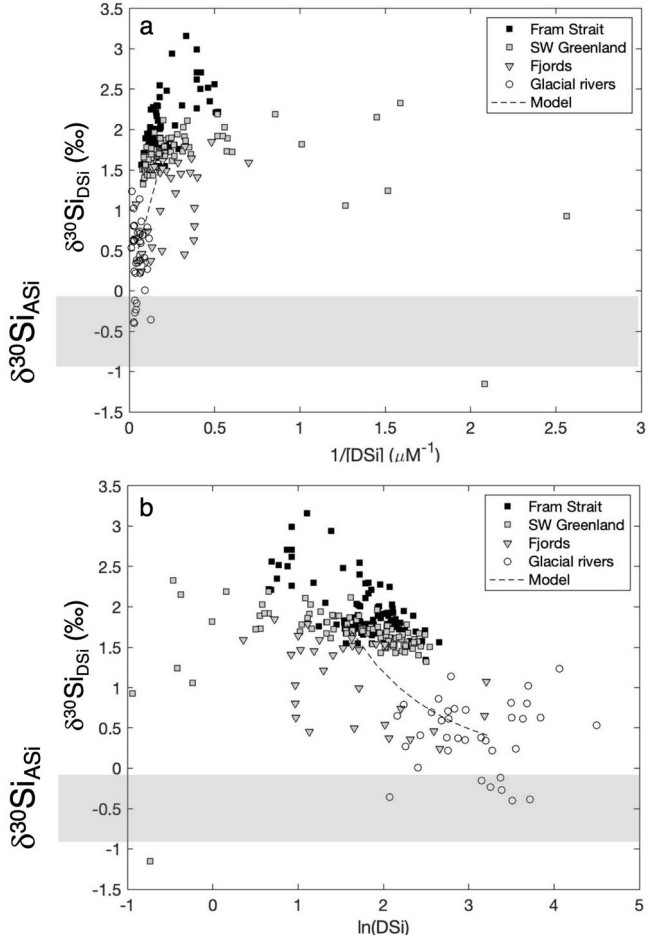

**Fig. 3 | Stable silicon isotopic composition of natural waters and amorphous silica from the Arctic.** Seawater $\delta^{30}Si_{DSi}$ from the Fram Strait[74], SW Greenland between Narsarsuaq and Nuuk regions[53], the fjord system near Nuuk (Ameralik fjord and Nuup Kangerlua)[13] (See Fig. 1), and SW Greenlandic glacial rivers[8,10] against (**a**) the inverse of the dissolved silicon concentration ([DSi]); **b** natural logarithm of [DSi], where the gradient is the assumed fractionation during Si uptake under closed conditions. Dashed lines show a simple mixing model between a freshwater source ([DSi] = 25 μM; $\delta^{30}Si_{DSi}$ = +0.41‰), which are mean values from[8,10], and a marine source ([DSi] = 4.8 μM; $\delta^{30}Si_{DSi}$ = +1.78‰) from[53]. Grey bars show the range of $\delta^{30}Si_{ASi}$ from across the Arctic. Note that while DSi concentrations are generally measured colorimetrically (i.e., "truly dissolved" silicic acid), $\delta^{30}Si_{DSi}$ is generally measured on filtered samples (commonly either 0.2 or 0.45 μm) using inductively-coupled plasma mass spectrometric methods, and care must be taken to ensure that there is no fine particulate matter present that influences the measurement[28].

settings[49]. A trend towards lighter isotopic compositions away from the glacier could be a result of a greater contribution of BSi dissolution. However, a mechanism is still required to explain the high $\delta^{30}Si$ in the glacier-proximal pore fluids: glacially-derived ASi is isotopically light, and the dissolution of BSi cannot drive pore fluids heavier than the overlying water. The increase in asymptotic DSi concentrations coupled with a trend towards lighter isotopic compositions with distance away from a glacier is consistent with a removal process (preferentially removing the lighter isotopes) within the freshest glacially-sourced sediments that are deposited rapidly within a fjord. One possible mechanism is that authigenic precipitation of secondary minerals take up DSi in addition to metal cations, often termed reverse weathering[41]. An alternative mechanism relates to the supply of reactive metal minerals—largely iron (Fe) and manganese (Mn) oxyhydroxides—from subglacial

weathering: DSi adsorption onto these Fe/Mn oxyhydroxide surfaces will result in a strong fractionation that leaves the heavier Si isotopes in the dissolved phase[15,46,49]. Whilst such reactive iron phases are found in sediments worldwide, including in glaciated sediments (e.g., ref. 50), the flux of these particles will be greater closer to the glacier within fjords[51], which would provide more adsorption surfaces for DSi. New sediment sequential extraction methods, combined with stable isotope and elemental analyses, are beginning to shed light on different operational pools of reactive silica in marine sediments, including a likely component of poorly-crystalline authigenic silica and adsorbed Si in the weak acid-extractable fractions (see Supplementary Notes and Fig. S1)[15,43,49]. Whilst the adsorption of DSi may not have a substantial role in long-term Si burial, per se, it could play a role in pre-concentration of Si onto nucleation sites for the formation of authigenic phases—the supposition of such a continuum between adsorption and pre-cipitation processes at least forms a testable hypothesis for future research. Regardless of any link with authigenesis, the DSi adsorption (and potential later desorption or release during microbial reduction of Fe-Mn oxyhydr-oxides e.g., ref. 52), will be important in the dynamic cycling of DSi within the fjord and coastal region. Importantly, given the lower pore fluid DSi con-centrations, diffusive (and likely advective) fluxes will be lower closer to glaciers relative to the coastal shelf sediments[49].

Benthic Si cycling in glaciated fjords and coastal settings can be com-plex and highly heterogeneous with large gradients in sedimentation rates and bottom water conditions, especially when influenced strongly by resuspension due to physical processes such as tidal mixing and iceberg scour. A lot more work is needed to fully quantify and understand the seafloor contribution to the supply of nutrients to coastal and open ocean ecosystems. Not only is there a large spread (∼ 0.01 to 10 mmol m$^{-2}$ day$^{-1}$ [15]) in benthic DSi fluxes in fjords and coastal shelf sediments (Fig. 4c), but also a variation in pore fluid isotopic composition due to different silica sources and sinks. For example, core incubation data off Southern Greenland revealed high fluxes of isotopically light DSi ($\delta^{30}Si$ trending towards 0‰ after one day of incubation), which was likely due to dissolution of benthic sponge BSi[48]. Novel geochemical methods, including observational and model-based approaches, are going to be crucial in disentangling these different sources and sinks.

## Continental shelf and beyond

While there are clear links between subglacial weathering, fjords, and coastal pelagic and benthic systems, some key questions remain unanswered. To what extent are fjord and coastal ecosystems a true continuum? Is the glacially-derived DSi that reaches the ocean important for coastal ecosys-tems? The answer to these questions will depend on the ecosystem structure, geography, and the factors limiting biological production in shelf waters.

Seawater $^{32}Si$ incubations off Southwest Greenland show that diatom communities are limited by Si to varying degrees along the continental shelf. At the southernmost part of Greenland, near Narsarsuaq and Cape Farewell, the diatom communities show no evidence of kinetic Si limitation (i.e., Si uptake by diatoms is suboptimal due to low DSi). However, further north (along the West Greenland Current, near Nuuk) there is evidence for at least seasonal kinetic limitation in continental shelf, slope and off-shelf waters during summer, and in the fjords near Nuuk[32]. Compared to the south, the on-shelf waters further north are characterised by lower DSi concentrations, higher turbidity and Chl *a*, and an exceptionally high ratio of (ASi + BSi) to DSi. The average for this ratio [(ASi+BSi)]/[DSi] off Nuuk is 119 ± 14% (*n* = 50)[53], over an order of magnitude higher than Narsaq, and four-fold higher than the maximum observed in open ocean gyre conditions[54]. Fur-thermore, there is a shift in the relationship between coastal [DSi] and $\delta^{30}Si_{DSi}$ along the Greenlandic boundary current, indicative of lower apparent isotopic fractionation (Fig. 3a, b)[53]. This trend could reflect changes in water mass distribution along the path of the West Greenland Boundary Current, with an increase in Coastal Water towards the north[55] and an associated increase in detrital silica input[12].

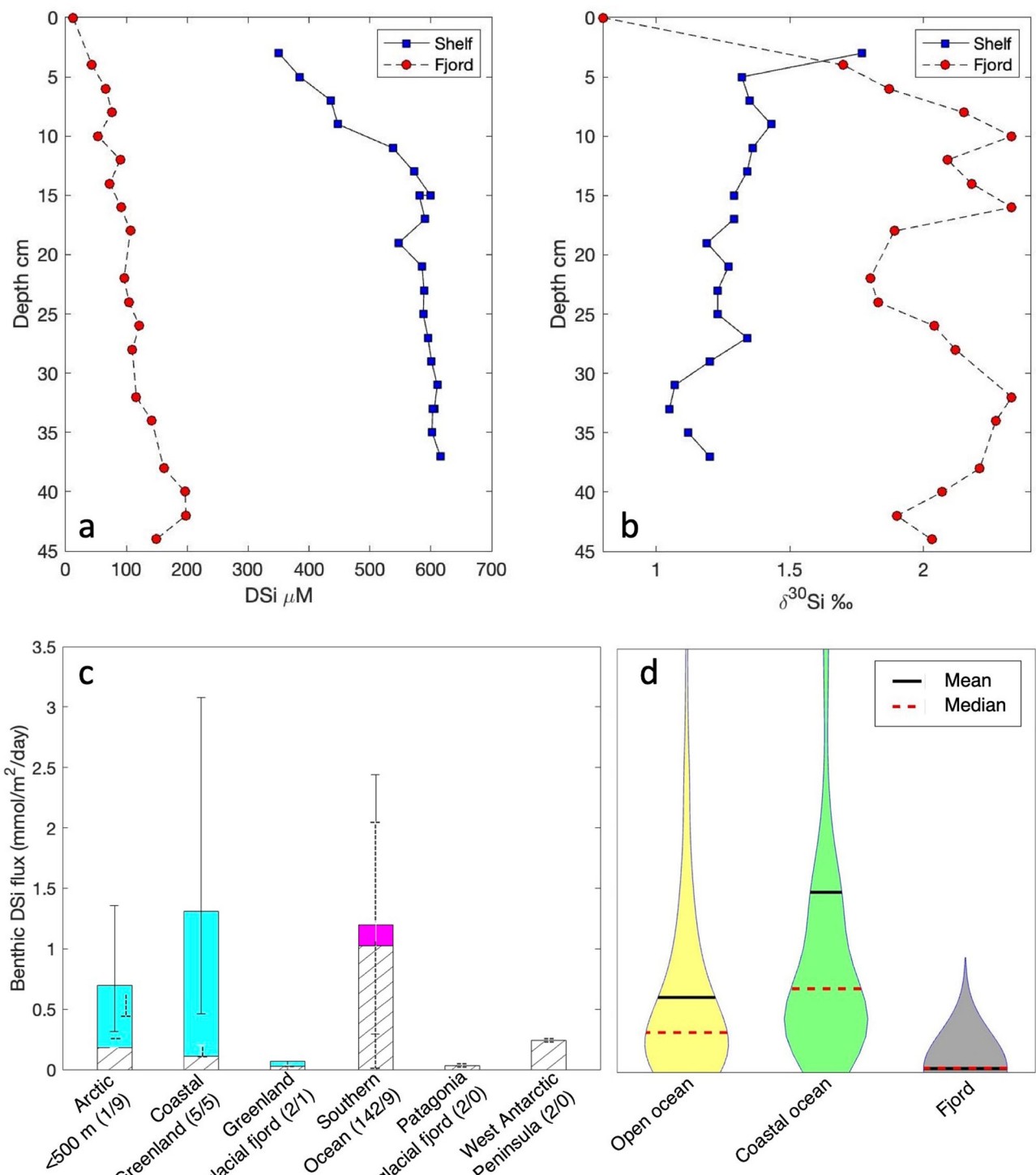

**Fig. 4 | Pore fluid silicon concentrations and stable isotopic compositions from Greenland, and a global compilation of benthic dissolved silicon fluxes. a** Pore fluid chemistry [DSi] and (**b**) stable Si isotope composition for a shelf location in the Godthåb Trough, near Nuuk[48] and within Nuup Kangerlua fjord, Southwest Greenland[49]. **c** Compilation of benthic DSi fluxes from porewater profiles (i.e., diffusive only fluxes, hatched bars) and core incubations (solid bars) from the Arctic margins[75,76], coastal Southwest Greenland[48], glacial fjords from Greenland[49], Southern Ocean (compiled by ref. 48), glacial fjords from Patagonia[15], and West Antarctic Peninsula[46]. The numbers in parentheses are the number of cores used for porewater profiles/core incubations; bars show range of values for each location (unless only one observation has been made, in which case the bars show propagated errors). See Fig. 1 for locations. **d** Global compilation of benthic DSi fluxes based on diffusive pore fluid profiles from open ocean, coastal and fjord sites (compiled by ref. 15).

What is driving the fundamental downstream change in silicon cycling around the coast of Greenland, from the East Greenland Current into the West Greenland Current (EGC and WGC respectively), and from Narsaq to Nuuk? The answer could lie in the response of the diatom community to the supply of meltwater into the Arctic freshwater budget specifically from the Greenland Ice Sheet (GrIS). Meltwater input from the eastern GrIS only currently starts to contribute significantly to the freshwater pathways in the EGC south of 65°N, according to model data, and meltwater from the

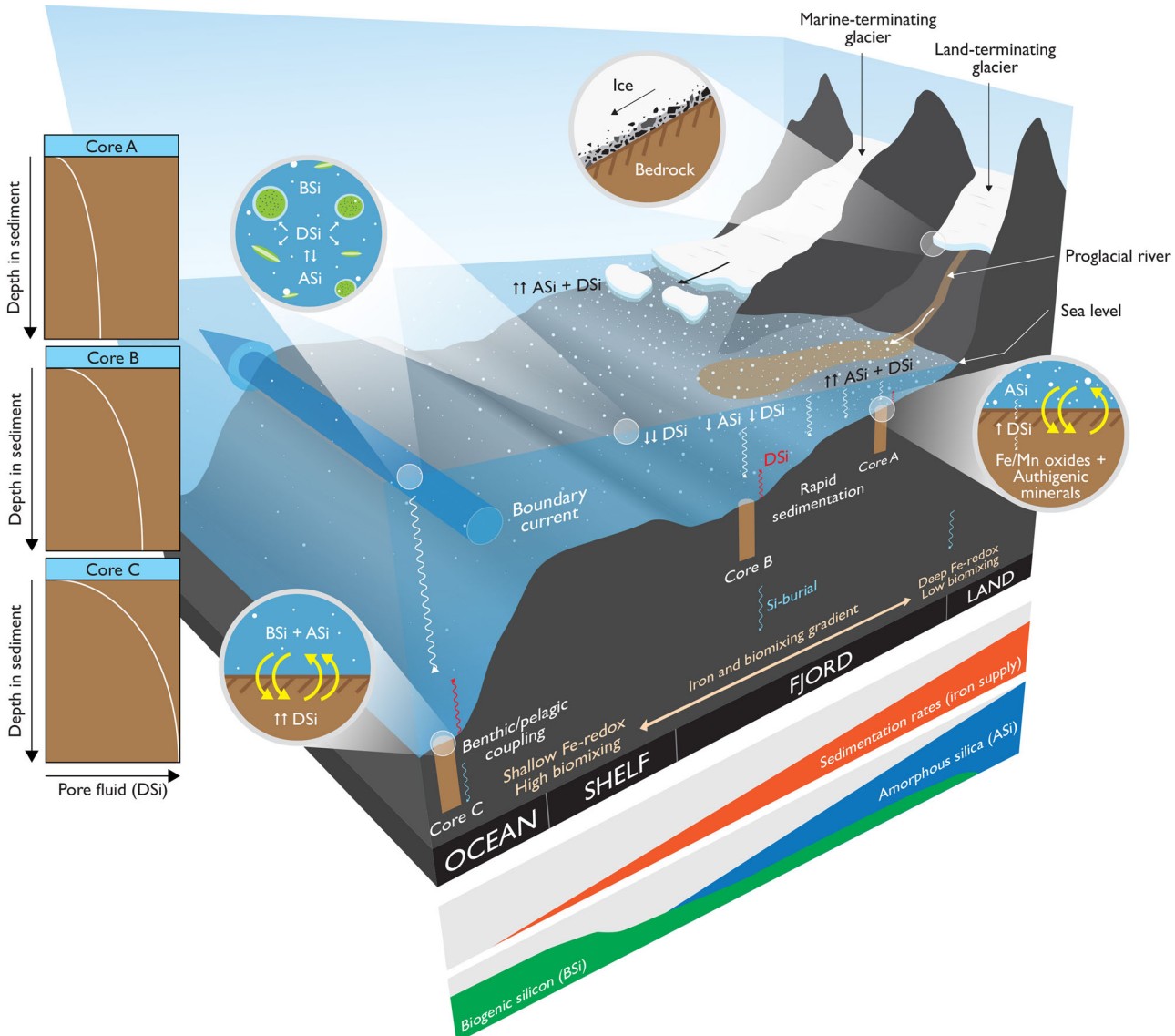

**Fig. 5 | Summary of silicon cycling of Southwest Greenland.** Subglacial weathering releases reactive particulate silica and dissolved silicon with a light isotopic composition. Glacial physical weathering products largely settle out of the water column; diagenetic reactions within the fjord sediment take up DSi, resulting in a relatively low benthic diffusive flux of DSi. Glacial particles that can escape the fjord supply reactive detrital material that can support diatom growth. Relatively high pore fluid [DSi] in coastal sediments results in a higher benthic diffusive flux of DSi into overlying waters. Virtual coring sites used in the modelling of the hypothetical fjord system are shown (proximal (A), mid-fjord (B), and distal (C) from fjord head to fjord opening) with pore fluid [DSi] profiles shown in side panels; key boundary conditions and assumptions are shown under the schematic. Image by Jamie Oliver, British Antarctic Survey.

western GrIS dominates input along the southwest coast (e.g., ref. 56). Furthermore, there is strong eddy-driven mixing at the southern tip of Greenland, whereas the waters off Nuuk are more stable and the surface Coastal Water (CW) is more dominant[55]. The difference between the eastern (Fram Strait) and western (Narsaq to Nuuk) Si cycling could arise if the input of GrIS meltwater, glacially-derived material and associated detritus is focused downstream of Southeast Greenland. Observations also support a continual input of terrestrial material along the Southwest Greenland coast. Multiple sources of evidence, including uranium-series radiogenic isotopes and glider bio-optical sensor data, indicate that lithogenic and other terrestrial material is added to the coastal surface waters along the length of the stretch of Southwest Greenland coast between Narsaq and Nuuk[12,57], where there is also a shift in DSi limitation among diatom communities[12,53]. As such, the addition of detrital material from mixed glacial sources will influence the growth of diatoms heterogeneously, depending on whether the fluxed material is freshly supplied and the nutrient limitation state of the diatoms. Further factors could also influence

the link between detrital phases and biological production, including the impact of particle (e.g., ASi) dissolution on the cycling of other elements and, in turn, seawater elemental stoichiometry, phytoplankton community composition (e.g., diatom vs. *Phaeocystis* dominated) and carbon export.

A picture is emerging of a "conveyor belt" of particulate silica dissolution that connects diatom production inshore and offshore, with strong uptake of DSi maintaining low [DSi] concentrations in continental shelf waters off SW Greenland (Fig. 5). Biogenic silica production is high, but a balance of reduced dissolution of fresh diatom BSi and relatively high export are needed to maintain low ambient [DSi]. Low dissolution within surface waters would be driven by low temperatures, but also strong export of diatom opal, most likely facilitated by aggregation processes[32,58], with remineralisation of the opal occurring deeper in the water column and/or at the seafloor[12,48]. The slow dissolution of any remaining (ASi + BSi) in the water column, both glacially-derived non-biological material and dead diatom detritus, could help to fuel summer production of BSi offshore despite low ambient [DSi]. Our estimates

**Table 1 | Benthic fluxes calculated from the reaction transport model thought experiment**

| Core | | | Total | Diffusion | Bioturbation | Advection | Bioirrigation |
|---|---|---|---|---|---|---|---|
| A | Fjord head | $J_{tot}$-DSi | 79.3 | 37.0 | 14.9 | −0.1 | 27.5 |
| B | Fjord mouth | $J_{tot}$-DSi | 133.6 | 57.5 | 23.1 | −0.02 | 53.0 |
| C | Shelf | $J_{tot}$-DSi | 276.7 | 127.1 | 51.2 | −0.01 | 98.4 |

Benthic fluxes ($J_{tot}$-DSi in $\mu mol\ cm^{-2}\ year^{-1}$). See Fig. 5 for relative location of cores.

suggest that over two-thirds—and potentially all—of the BSi production at the Nuuk stations could be sustained in summer by dissolution of these non-diatom silica sources, depending on the dissolution rate of the detrital material used[53].

As a continuation of the fjordic environment, benthic fluxes are likely to be an important source of DSi in these glaciated shelf and off-shelf regions, perhaps even more so given the generally higher pore fluid DSi concentrations found away from the glaciers themselves (Fig. 4a, b)[15,48]. Some estimates of these benthic fluxes are a similar order of magnitude to the water column BSi production. For example, benthic DSi fluxes off southwest Greenland calculated from sediment core incubations (Fig. 4c), are the same order of magnitude as integrated BSi production in the overlying water column estimated from $^{32}Si$ seawater incubations (0.13–14.4 mmol Si $m^{-2}$ $d^{-1}$)[12] and have the isotopic fingerprint of dissolving BSi[48]. This observation points towards strong benthic-pelagic cycling in these coastal environments, with benthic fluxes contributing to the support of biogenic production. Upwelling and mixing (by internal waves, tides, storms, currents, and other instabilities) in high-energy coastal environments will act to entrain DSi released by benthic processes into the near surface for utilisation by diatoms.

## Breakthroughs driven by methodological advances
### Combining isotope systems in early diagenesis studies
Our initial interpretation of the heavy silicon isotopic composition of the pore fluids led to a clear hypothesis that there is a redox-sensitive coupling between Si and iron (Fe) in glaciated environments. A logical next step was to combine the Si isotope data with stable Fe isotopes, which are fractionated by reductive and non-reductive dissolution processes, to provide additional evidence for linkages between these two nutrient systems. To date, we have applied this combined approach to pore fluids from fjord sediments in Patagonia[15]. The findings revealed a high diffusive flux of dissolved Fe from glaciated fjord sediments, in contrast to a relatively low Si flux, driven by both reductive and non-reductive dissolution of glacially-sourced reactive Fe phases, reflected by a wide range of pore fluid stable Fe isotopes ($\delta^{56}Fe$ from −2.7 to +0.8‰). The downcore trends in pore fluid isotopic compositions indeed supported the removal of dissolved Si by oxidised iron phases under certain redox conditions. Specifically, the lowest porewater $\delta^{56}Fe$ compositions, likely driven by Fe microbial reduction, intersect with the highest $\delta^{30}Si$, implying Si adsorption onto Fe oxyhydroxides. Si adsorption to freshly precipitated Fe oxyhydroxides, together with high sedimentation rates, contribute to the low diffusive flux of Si out of the sediments[15].

### Disentangling and quantifying Benthic Si cycling through reaction-transport modelling
Biogeochemical modelling is required to unravel the complex diagenetic reactions occurring in the fjord water column and sediments, to quantify fully the role of particle-water interactions in Si cycling[40,59,60]. The Biogeochemical Reaction Network Simulator (BRNS) is an example of a biogeochemical model used to critically quantify and better understand Si cycling in the benthic realm (as well as seafloor [in]organic carbon, metal, and nutrient dynamics)[61]. BRNS is a flexible simulation environment designed for mixed kinetic-equilibrium reaction systems[62,63], which simulates the evolution of solids and solute concentration depth profiles in porous media[64,65]. The model resolves the main transport processes,

i.e., sediment and pore fluid advection, diffusion of solutes, and bio-mixing of solids and solutes[66]. Over the past years, we developed a reaction network that allows the simulation of benthic silicon cycling, including Si isotope dynamics (Si-BRNS[14,15,46]). In brief, the reaction network calculates changes in DSi through the dissolution of BSi, precipitation of authigenic silica (AuSi)[46] (analogous to early reverse weathering products), and the dissolution of amorphous (and other lithogenic) silica (ASi/LSi)[14]. It also accounts for (an implicit) redox-driven DSi adsorption/desorption onto iron oxyhydroxides[14,15]. These processes can be resolved for $^{28}Si$ and $^{30}Si$, and thus the model also traces the evolution of $\delta^{30}Si_{DSi}$[14,46]. The Si-BRNS is forced by overlying bottom water conditions, such as the concentration and isotopic signature of DSi, the temperature and salinity of the bottom waters, sedimentation rates and biomixing parameters, the deposition fluxes of ASi and BSi, which in turn are a function of the magnitude of glacial input (ASi) and local pelagic primary productivity (BSi), respectively (see Supplementary Methods and Tables S1–S3)[15].

We constructed a thought experiment to start illustrating the sensitivity of the earlier discussed diagenetic process interplay that controls benthic Si cycling in fjord systems to along-fjord gradients in bottom water conditions and sedimentation rates, which will be critical in determining the residence time of particles at the sediment-water interface. To this end, we set up a reaction-transport model that explicitly resolves transport as well as the diagenetic reaction processes that have been identified by observational data and previous modelling efforts as important controls on benthic Si cycling and fjords (discussed above). Along a hypothetical fjord land-to-ocean gradient (Fig. 5), we then ran a set of steady-state simulations at three virtual "coring sites": proximal, middle and distal, from fjord head to fjord opening. We compiled literature data on the range of bottom water boundary conditions linked with Si and Fe cycling, and made some basic assumptions regarding sedimentation rates and diagenetic rate constants (dissolution, adsorption, and precipitation) based on previous modelling efforts (see Supplementary Methods, Tables S1–S3). Model results reproduce the widely observed spatial trend in DSi profiles—with asymptotic concentrations increasing away from the glacier—given our along-fjord boundary conditions, model parameters and diagenetic reaction network. They also show that we need to invoke DSi adsorption to reactive metal phases to capture the field isotopic observations fully. Importantly, simulated benthic Si fluxes increase away from the glacier (Table 1), in line with observations (Fig. 4c). Model results reveal that this increase is due to an increase in BSi dissolution. Whilst our Gedankenexperiment shows that the proposed mechanism for how early diagenetic processes drive benthic Si cycling in these environments can indeed explain the observed trends, it also highlights the sensitivity of the system to along-fjord and thus also temporal changes in fluxes, bottom water conditions, residence times and reaction kinetics, and that further research is required to constrain these unknowns.

## Outlook
### How important is the cycling of glacially-derived silicon on a regional to global scale?
Despite being important to the localised ecosystems in Arctic coastal waters where the concentration of DSi is low, it could be argued that the glacial

supply is negligible for the supply of silicon to the global ocean. We do not have a good constraint on the spatial extent of the trapping of DSi in glaciated coastal environments. Furthermore, the impact of glacially-sourced DSi on productivity is likely going to differ between the Arctic and the Antarctic, given the latter generally experiences higher background seawater [DSi] in surface waters due to mixing with Si-rich Circumpolar Deep Water. However, in addition to large areas of the Arctic (e.g., Svalbard[33]) and subarctic[32], there is evidence for at least seasonal silicon limitation in surface waters in part of the Southern Ocean[67], and around some subantarctic islands (e.g., Scotia Sea[68]), largely due to diatom uptake. As such, any supply that fuels considerable diatom uptake of silicon could contribute to the overall budget of Si in the ocean even if it is not seen in the dissolved fraction. Critically, this silicon input is happening right where it is needed, in a low DSi region that experiences Si limitation. More work is required to determine whether this silicon fertilisation is happening in other regions, and whether it has an impact in regions of the Arctic experiencing higher background DSi concentrations, e.g., Arctic Canada and Baffin Bay. The potential trapping of Si by reactive Fe and Mn-bearing phases could also impact the bioavailability of these important micronutrients, and further investigation is required into the impact of these processes in areas where algal growth is limited or co-limited by trace metal availability.

## How will Si cycling in glaciated environments change in a warming world?

Multiple factors influencing the cycling of silicon in glaciated environments are highly likely to be impacted by ongoing warming in the polar regions[69]. The nature and extent of glacial weathering will change, as will the interaction between glaciers and fjord water, with grounding lines retreat and, eventually, the transition of marine-terminating glacier to land-terminating glaciers. Furthermore, a longer growth season could impact nutrient limitation and the role of benthic recycling. However, there are several outstanding questions surrounding the mechanisms and influences of Si cycling within glaciers themselves that—at this stage—severely limit the confidence we can place on future predictions, including the role of supraglacial processes, or the connectivity, lithology and mineralogy of the sub-glacial environment. There are also ongoing compounding and multiple stressors acting on important biological and abiological processes across the land-ocean continuum, which are likely to differ between the Arctic and the Antarctic. For example, the temperature and nutrient content of Arctic seawater is likely to change with continued atmospheric warming and the increase in marine input from the lower latitudes ("borealisation")[70]. Whilst both water column and benthic reactions are likely to depend on temperature and the availability of organic matter and other nutrients (such as iron and phosphate), these links have yet to be quantified.

## What methods do we need to address these questions?

The land-ocean continuum is a complex environment characterised by dynamic cycling over a range of spatial and temporal scales, especially in the case of glaciated margins where there is the interaction between multiple water masses and reactive particulate phases. Multiple approaches are needed to disentangle this complexity. Autonomous systems provide the means to observe the system at higher temporal and spatial resolutions than previously possible using traditional research ship or small boat-based approaches. Autonomous underwater vehicles, such as gliders, can measure both physical and bio-optical properties of the water, allowing for the tracing of meltwaters—as well as associated particulates and dissolved constituents—out into coastal currents[57]. The deployment of multiple vehicles, especially with lower cost autonomous vehicles becoming increasingly available, will improve our four-dimensional view of glaciated land-ocean interfaces further. Novel biogeochemical sensors are also becoming commercially available that allow for robust, high spatial and temporal resolution measurement of nutrients, including DSi, and are already being tested in the challenging environments of glaciated fjords. Microfluidic, or lab-on-chip, DSi sensors deployed on a mooring have successfully captured variability in DSi and nitrate concentration over a summer season in Nuup Kangerlua; when paired with CTD measurements over the same resolution, these observations provide unprecedented insight into the shifting sources (marine vs. freshwater) of these two critical nutrients[71]. We need a better understanding of the role of resuspension processes in releasing nutrients sourced from glacial sediments, not captured by core incubations or benthic lander platforms (e.g., resuspension facilitated from stochastic atmospheric events, bioturbating megafauna, etc.). Such sensors could be the key to developing benthic lander technology and in situ observations of seafloor processes, in combination with other isotope geochemistry methods (e.g., radioactive radium isotopes[72]). Finally, there is a critical need to progress incubation experimentation for testing both the microbial response to nutrient supply from glacial particulate sources on a range of spatial and temporal scales, from individual cells to populations in the water column and seafloor sediments. These experiments could include combining bio-geochemical (especially rate) measurements with 'omics' methodologies, which can identify the molecular pathways underlying these responses. In addition to observational data, the development of extraction methods and reaction transport modelling tools that are capable of integrating the myriad of processes impacting Si cycling in glacial environments is needed, and could benefit from machine learning innovations. The results discussed in this synthesis highlight the importance of biogenic Si cycling, Fe(oxyhydr) oxide sorption of Si, deposition and diffusion, and interacting (in)organic processes, and model-based sensitivity studies, together with data-model integration, could be utilised to interpolate to system-wide scales, test hypothesised dynamics and processes, and predict future changes robustly.

## Data availability

All data presented in this paper are available from PANGAEA: Glider data https://doi.pangaea.de/10.1594/PANGAEA.931292. Bottle data https://doi.pangaea.de/10.1594/PANGAEA.907905. Pore fluid data https://doi.pangaea.de/10.1594/PANGAEA.907905. Fjord data https://doi.pangaea.de/10.1594/PANGAEA.930217. Arctic rivers https://doi.pangaea.de/10.1594/PANGAEA.902194. Core incubations https://doi.pangaea.de/10.1594/PANGAEA.907904. Or in the Supplementary Dataset.

## Code availability

Model code and results are available from Zenodo: https://zenodo.org/records/14794831.

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

## Acknowledgements

K.H., H.C.N., F.S.d.F. received funding from the European Research Council (Grant Agreement ERC- StG-ICY-LAB-678371); K.H., J.E.H., A.B. received funding from the Royal Society (Grants RGF\EA\181036 and URF\R \180021); K.H., J.H., H.P., J.W. received funding from NERC-CONICYT project PISCES (NE/P003133/1-PII20150106); K.H., S.A., F.S.d.F. received funding from UKRI-NERC ChAOS (NE/P006493/1); K.H., H.P. and R.J. are also supported by the BIOPOLE National Capability Multicentre Round 2 funding from the Natural Environment Research Council (grant no. NE/W004933/1) and SiCLING (grant no. NE/X014819/1); L.F. is supported by NERC GW4 + DTP; J.W. and S.T. are supported by the Research Council of Norway through its Centres of Excellence funding scheme, project number 332635 (iC3: Centre for ice, Cryosphere, Carbon and Climate) and Research Council of Norway project no 334596; T.W. is supported by the China Scholarship Council; S.v.d.V., F.S.d.F. and S.A. acknowledge funding from BELSPO (project DEHEAT); J.H. acknowledges funding from NSF (project MEGA award no. 2232980). All authors are thankful to the Captain, crew and scientists on board the R/V *Belgica* during the DEHEAT campaign in for support in collecting samples in 2023. With thanks to J Oliver (BAS) for Fig. 5.

## Author contributions

K.H., E.M.S.W. devised the manuscript; K.H. wrote the first draft and produced Figs. 3, 4, Supplementary Notes and Fig. S1; H.P. provided Fig. 1; H.P., R.J. provided new data; S.T. provided Fig. 2; F.S.d.F. carried out the modelling experiment, and wrote Supplementary Methods, Tables S1–S3, and provided Supplementary Fig. S2; all authors (K.H., F.S.d.F., S.A., A.B., L.F., J.E.H., J.R.H., R.J., J.K., L.M., H.C.N., H.P., S.T., S.v.d.V., J.W., T.W., E.N.S.W.) participated in original research, discussion and manuscript editing.

## Competing interests

The authors declare no competing interests.

## Additional information

[1]British Antarctic Survey, High Cross, Cambridge, UK. [2]School of Earth Sciences, University of Bristol, Bristol, UK. [3]BGeosys, Department of Geosciences, Université libre de Bruxelles, Brussels, Belgium. [4]iC3: Centre for ice, Cryosphere, Carbon and Climate, Department of Geosciences, UiT The Arctic University of Norway, Tromsø, Norway. [5]National Oceanography Centre Southampton, Southampton, UK. [6]Department of Ecology, Faculty of Science, Charles University, Prague 2, Czechia. [7]UK Centre for Ecology & Hydrology, Environment Centre Wales, Bangor, UK. [8]Department of Earth and Environmental Science, University of Pennsylvania, Philadelphia, PA, USA. [9]Dauphin Island Sea Lab, Dauphin Island, AL, USA. [10]Stokes School of Marine and Environmental Sciences, University of South Alabama, Mobile, AL, USA. [11]Greenland Climate Research Centre, Greenland Institute of Natural Resources, Nuuk, Greenland. [12]Department of Estuarine and Delta Systems, Royal Netherlands Institute for Sea Research, Yerseke, The Netherlands. [13]Department of Earth Sciences, University of Cambridge, Cambridge, UK. [14]Department of Biology, University of Antwerp, Wilrijk, Belgium. [15]Department of Marine Science, University of Otago, Dunedin, New Zealand. [16]National Institute of Water and Atmospheric Research, Wellington, New Zealand. [17]Plymouth Marine Laboratory, Prospect Place, The Hoe, Plymouth, UK.
✉e-mail: kathen@bas.ac.uk

