## [Peer Review file · Communications Earth & Environment]

Insights into silicon cycling from ice sheet to coastal ocean from isotope geochemistry

Corresponding Author: Professor Kate Hendry

Version 0:

Decision Letter:

Dear Professor Hendry,

Your manuscript titled "Silicon cycling from ice-sheet to coastal ocean: insights from isotope geochemistry" has now been seen by our reviewers, whose comments appear below. In light of their advice we are delighted to say that we are happy, in principle, to publish a suitably revised version in Communications Earth & Environment, provided you better clarify how temperature and nutrient shifts impact Arctic and Antarctic environments, soften and caveat your claims of the role of silicic acid in diatom growth to account for the concerns of Reviewer #1, and clearly define the separation between fjord and continental margin, as well as the mixing zone between seawater and freshwater.

We therefore invite you to revise your paper one last time to address the remaining concerns of our reviewers. At the same time we ask that you edit your manuscript to comply with our format requirements and to maximise the accessibility and therefore the impact of your work.

EDITORIAL REQUESTS:

*****Please take care to match our formatting and policy requirements. We will check revised manuscript and return manuscripts that do not comply. Such requests will lead to delays. *****

SUBMISSION INFORMATION:

OPEN ACCESS:

Communications Earth & Environment is a fully open access journal. Articles are made freely accessible on publication. For further information about article processing charges, open access funding, and advice and support from Nature Research, please visit <https://www.nature.com/commsenv/open-access>

At acceptance, you will be provided with instructions for completing the open access licence agreement on behalf of all authors. This grants us the necessary permissions to publish your paper. Additionally, you will be asked to declare that all

required third party permissions have been obtained, and to provide billing information in order to pay the article-processing charge (APC).

Link Redacted

Best regards,

Alireza Bahadori, PhD
Associate Editor
Communications Earth & Environment

REVIEWERS' COMMENTS:

Reviewer #1 (Remarks to the Author):

Glacial weathering has been shown to deliver inputs of dissolved and amorphous silica to coastal waters at regional scale. These inputs of Si are significant at the global ocean scale.

The dynamics of Si in the ecosystems adjacent to the ice-sheet is actually complicated because it is controlled by an ensemble of physical, biogeochemical, and geochemical processes that occur both in the liquid and solid phases, as well as at the sediment-water interface. Using data describing typical systems of the two hemispheres located in Greenland, Iceland, Svalbard, Norway, Alaska, Patagonia, and in the west Antarctic Peninsula, the aim of this manuscript is to shed light on this complexity. To reach this aim the authors take into account of isotopic tools. First, they show that a light isotopic signature characterises "glacial silica", both in dissolved and amorphous phases; this light isotopic signature is a kind of « chemical fingerprint ». For example $\delta^{30}\text{Si}$ provide a mechanistic understanding of subglacial amorphous silica formation in dissolution in the geo-ecosystem of the land-ocean continuum. Second, the authors combine the use of stable isotopes and radioactive ^{32}Si , and reaction-transport to quantitatively constrain the processes at work in this complicated environments and to simulate the cycling of Si along the land-ocean continuum.

This review article confirm evidences that amorphous silica is rapidly mobilisable through dissolution in seawater to release silicic acid. However, the role of amorphous silica dissolution in fjord Si cycling remains unquantified. It recalls increasing evidences of dynamic Si cycling during early diagenesis in high latitude sediments. It shows the importance of adsorption of silicic acid onto Fe/Mn oxyhydroxydes surfaces and of Si-Fe interactions at redox boundaries.

Interestingly, benthic Si cycling through reaction-transport is disentangled and quantified using a "Biogeochemical Reaction Network Simulator". A synthetic scheme showing Si cycling in the SW Greenland coast is very helpful to illustrate the processes at work in this typical glacial geo-ecosystem.

Concerns:

1-This manucript evokes a possible evolution of Si cycling in glaciated environments submitted to climate change. This is a crucial point. Unfortunately, the present version is too vague on this crucial point, as far as the potential impacts of changes in temperature and nutrients are concerned. They might be different for the arctic and antarctic environments.

2-That silicic acid delivered to coastal water through the Si glacial weathering can support a "substantial proportion of diatom growth" is mentioned in the abstract. Clearly, this is to be demonstrated, particularly for the Southern Ocean where much of the silicic acid input to surface waters comes from below, i.e, from upwelling of silicic acid rich deep water.

Reviewer #2 (Remarks to the Author):

Hendry and colleagues present a comprehensive review of Si behavior in glaciated margin systems with an intention of highlighting the insights related to benthic fluxes that can be gleaned from silicon isotopes specifically. As part of the review, the authors provide a synthetic on which they perform a sort of sensitivity analysis that illustrates benthic Si fluxes along an offshore gradient.

Overall, I think this would be a useful paper for the community and I appreciate the idea of putting it all together. My main overarching comment is that I was not entirely clear on what NEW insights stem specifically from isotopes and what come from the sequential extraction based concentration measurements and what come from adding a . If the goal is to highlight isotopes specifically, it may have missed. My impression was that it provides a more holistic picture of the near shore dynamics on glaciated margins with the measurements and models together.

I also list editorial comments.

line 105-106... dissolution incubations recommended for light-dark cycles and T differences... presumably these are meant to understand biological activity... but I wonder about the light component. In sediment, if they are buried shallow enough to receive light, only the thin layer of the uppermost sediment would be exposed? Is the hypothesis that this boundary layer hosts benthic diatoms that might accelerate A Si dissolution?

Figure 2 caption..."See Supplementary Information for reactive silica (ASi) sequential extraction data:: Maybe elaborate somewhere about this because I do not know why I am being directed to look for it.

line 114: Box one discusses silicon isotope values or measurements, add either word after isotope.

Throughout the ms discussion of "light" isotopic signals would be more accurately portrayed as low values... as that is what is measured. I know this may seem pedantic but review articles are broadly read and adhering to strictly correct terminology is useful for linking this work to past and future work.

line 116-117: Something seems off here: Glacial<non-glacial so parenthetical text is backwards?

line 119-120: "appears to change over a melt season", how does it change (e.g. increased or decreased values with increased or decreased DSi?) and what does connectivity mean?

Line 127: Signature should refer to an endmember: Replace light isotopic signature with low isotope values since $\delta^{30}\text{Si}$ of glacial silica, although low in value, varies.

Lines 135-139: While it is in the weeds, it seems like the size distribution matters in this case of crushing and re-crushing materials as well as Intense (longer) grinding mentioned on line 143 and for estimating something like reactive surface area either for modeling or experimental work.

line 150-151: I think this means $\delta^{30}\text{Si}$ values were linearly related to % glacial cover in the watershed?

Lines 160-163: Confusing because it seems like the authors are on one hand referring to silicate weathering related CO_2 drawdown as something that every reader understands so well that it does not need to be discussed or even referenced but yet at the same time they parenthetically state it is not well defined. Do they mean the process, the magnitude, what?

Line 173, remove the parentheses.

Line 181-182, delete text in parentheses: measurement nuance and not needed in a discussion of what forms are in fjord sediments/source DSi.

Line 193: I think it is more straightforward to say "why does DSi appear to behave conservatively in fjords?" because given the above paragraph talking about DSi as a major nutrient stimulating diatom growth, it just makes to sense to ask if it behaves conservatively.

Line 202: replace word location with "surrounding conditions" or something that describes more than just lat/long but things that may contribute to variations in nutrient supply, glacial proximity etc...

Line 221: "impacting the general inverse relationship", between what? given you have DSi increasing and ASi dissolving and Salinity varying. Be specific

Lines 236-240: Show predictions in Fig. 3 so reader can assess deviations from predicted.

Lines 265-267: What is the role of sediment accumulation rate on this relationship? the DSi in the upper 50 cm of shelf has longer to accumulate or bSi longer to dissolve.

Figure 4 caption: "for porewater profiles and core incubations respectively. Unclear if respectively is supposed to mean with respect to the slash. maybe write as cores used for porewater profiles/core incubations ?

Lines 398-403: I would recommend removing the mention of the Fe isotopes as they are not discussed again or here in detail and they do not add to the understanding that evidence exists for links between Si fluxes and dissolution/precipitation of Fe oxy hydroxides.

Line 434/Line 452. Models are thought experiments. No need to add language that makes it less clear what you did. Sensitivity analysis might be more useful than Gedankenexperiment.

Line 442: "reasonable" implies a value judgement, why not range of bottom water conditions.

Table 1: include something that characterizes the environment of cores A, B, C in table, e.g. fjord, shelf, slope.

Line 502: Methods to address questions: Can you combine this section with teh section above and make a table of methods/tools. Reading lists as text is difficult at any point but at the end of a paper it is nearly impossible. A table would be more valuable to readers.

Reviewer #3 (Remarks to the Author):

The pdf attached is the same text as below.

This paper presents an overview of recent advances on subglacial processes in the biogeochemical cycling of silicon. Many studies have been published in the last decade and it is indeed a 'hot' topic of interest to several scientific communities (climate / ocean / biogeochemistry / geosciences) because (i) it is relatively new (weathering in cold climates has often been overlooked because it was thought to be minor compared to humid and warm climates) and, (ii) also because of the dramatic changes these polar environments are currently facing due to global warming. So, this paper is welcome, and since most of the recent work has been published by this group in (too?) many different papers, the review really helps to get a much clearer overall picture of the processes. The paper is very interesting and is a nice and useful synthesis of these works.

However, I have some comments below to make the synthesis clearer and more appropriate. I can summarise my main concerns here: (i) homogenisation of nomenclature (e.g. silica vs. silicon; Si isotope notation; better separation/definition of processes and areas along the glacier-ocean continuum fjord/shelf/mixing....); (ii) more appropriate use of Si isotope models (by definition data from different Si isotope systems cannot fit the same isotopic trend); (iii) improve the synthetic Fig.5 so that it is more informative and it better reflects both the complexity of the processes discussed and the consequences for the

silicon (isotope) cycle. In this respect, some processes are not sufficiently discussed (e.g. reverse weathering and land processes); (iv) the methods on sediment leaching as well as the part would need some additional information in the supplementary material.

Damien Cardinal

Nomenclature and notation

- Silica vs silicon. Silica = SiO₂. Silicon = Si. This looks trivial but is too often incorrectly referred to, including in this paper. In the dissolved form, Si is under silicic acid and can be referred to as DSi. Sometimes this DSi comes from the dissolution of silica (biogenic, amorphous or even from quartz), sometimes not and comes from aluminosilicates (amorphous or crystalline) or from desorption, i.e. not from silica so the term dissolved silica is incorrect. Please do remind this (e.g. in the box) and try to respect it in this review (e.g. lines 82 and 146 silica should be replaced by silicon).

- Si isotopes (lines 330, 426, 546, box etc.). Avoid the notation d30DSi or d30ASi which is chemically confusing (superscript 30 associated with D or A and not Si is incorrect), and rather keep the notation d30SiDSi and d30SiASi (with DSi and ASi as subscripts) as this is the main notation used in this ms.

Different processes along the glacier - to - ocean continuum

- Lines 167 before the subsection on fjord. There is no mention on what can happen for land terminated glacier to the glacial material before it reaches the fjord / coast via the proglacial river. The presence of lakes for instance could dramatically change the particulate content downstream (settling); it might also change the DSi concentration and Si isotopic composition in case of freshwater diatoms Si uptake. Even without lake, glacial plains can be wide and under low slope forming meanders favoring settling in the proglacial rivers running at low speed. Has this aspect been studied in the different areas compiled here? Could it explain variability within land terminated glaciers as well as in between the 2 categories summarized in Fig. 5 (land-terminated or sea-terminated)? This is a well-known key property controlling the supply of particulate to the coast in non-glacial environments, why not for glacial environments?

- Section fjord vs. continental margin sediments. Similarly, it's unclear where is the separation all over the manuscript between fjord and continental margin. Is it the geomorphology as it appears since no other precision is given? However, what does matter is the mixing zone between SW and freshwater. Depending on the site, I assume this zone might be within the 'fjord' zone or as plume over the continental margin/shelf; of course, this also depends on type of glacier: land-terminated or sea-terminated. From the synthetic Fig. 5, the category is based only on this type of glacier: fjord is displayed as the interface between land and shelf but this part of the scheme does not look like a fjord and looks more like a plume extending into the sea. I would recommend to make a separation as 3 zones 100% freshwater, 100% seawater and the mixing zone in between; or clarify the choice / illustration / sections made here. This confusion extends to the third category 'coastal system and beyond' from line 310. The fig. 1, fig. 5 and the text should be organized accordingly for a better understanding of the authors' purpose.

- 280 and . I appreciate the discussion on the role of Fe-Mn oxide to adsorb significantly Si, but I miss a section on 'traditional' reverse weathering, i.e. formation of authigenic secondary minerals (sensu Michalopoulos & Aller, 2004) which was proposed on coastal sediments. It has been suggested to (partly) control also d30Si in open polar ocean sediment at low T°C (Closset et al., 2022). While this is briefly mentioned here, I see no reason why to favour the Fe-Mn and Si adsorption process compared to reverse weathering. In fact, more discussion appears at the end of the ms (from line 408) in the section on the Biogeochemical Reaction Network Simulator where it includes authigenic minerals formation and implicit representation of adsorption/desorption. This is confusing. I suggest to put the discussion on BRNS earlier when these processes are tackled.

- Fig. 5 is a pretty scheme, but is too general. It summarises too basic and trivial information, much simpler than the previous studies which are nicely discussed in this synthesis ms. I would suggest to complexify it to better reflect all these works by e.g. making different panels for different types of glacial environments (mixing zone, upstream fjord...) and/or adding d30Si variability and/or adding Si flux or Si contribution ranges from different processes as estimated from the (e.g. by pie chart).

Si isotopes models

- Fig.3 is not adequate to support the discussion and is at least partly misleading. In 3a and 3b, as reported, it is true that we expect a linear relationship for mixing (3a) or for diatom uptake in a closed system (3b). However, such a relationship would only be valid for a single isotopic system (as simply imposed by the isotopic mixing or Rayleigh equations), i.e. we should have linearity for each site, not only for each type of symbol, but also within a symbol such as glacial rivers or fjords where there are different sites. I acknowledge that glacial ASi d30Si could be considered the same everywhere (btw a key finding and very nice contribution of all these previous studies and well highlighted here), but DSi concentrations and d30Si-DSi of end members (for 3a) or DSi sources (for 3b) are highly variable (as are ASi concentrations). The scale makes these panels very busy, and they are not evidence of what is claimed, since potential linearity is not discussed per site, only rejected for the whole data, which is trivial and not relevant. Similarly, for 3c, each system should have its own mixing curve which is not shown. For the revised paper, one possibility could be to make 3 small panels for each site and put them in the supp mat, although I find the supp mat already extremely long.... The other possibility might be to just delete this figure without changing much of the text and just refer to the original articles.

- Fig. 4d See comment above on Fig. 3 there is no reason why all this data would plot on the same Rayleigh system; even within the same symbol type (e.g. just for Southern Ocean sites, the DSi concentration varies by a factor of 2 and d30Si by almost 1 pmil depending on the cores and bottom water DSi are order of magnitude higher than the ones in the Arctic, so the

PW isotopic data simply cannot plot on the same Rayleigh). Nothing can be expected from such plot with data from so different origins. Note that I do agree with the claim that heavier $d^{30}\text{Si}$ of PW in the proximal sites suggest preferential immobilization of light isotopes but again, this conclusion does not come from Fig. 4d in its current form.

Methodology in supplementary material

S1 text and table: Methodology is not the main topic of this paper, which I somewhat regret as there is no agreed method for measuring Si isotopes from coastal sediments, especially when high non-biogenic amorphous silicon is present. It should at least be highlighted here that several leaching/sieving methods are used and briefly discussed whether this could explain some of the variability found in different studies, e.g. in a table summarising these methods and then discussed. For example, why is Iceland so variable and higher (lithology is mentioned but this is not convincing)?

Model

This part is quite dense, and only described in the supp material. The idea is relevant and interesting and have been (partly?) published in a separated article. Since (i) it is not the core of this manuscript (ii) it is only shortly presented in the supp mat (iii) it's not my domain of expertise, I cannot fully evaluate this part. I do have some comments though:

- Clarify what is new in this part compared to Ward et al. 2022 (btw it's erroneously cited as Ward et al. 2020, while it's the 2022 in the reference list; please check) et Ng et al. 2022?
- I wonder if there are any data for the 4th assumption of the (supp mat: "Iron-adsorbed silicon (FeSi) occurs as a pre-formed silicon phase in the water column."). Is it an important feature of output? Please provide some reference / justification.
- With reference to my previous comments on the different categories chosen, how are the compositions of these cores affected by the location of the mixing between SW and freshwater? Is core B implicitly under the mixing zone?
- Would it be possible to plot real data on fig. S2 to validate the output??

Minor comments

94 Fig S2 is cited instead if Fig. S1.

160-162, needs to be developed since the link is not trivial. Is $\text{CO}_2/\text{H}_2\text{CO}_3$ involved in this weathering? Could it be only mechanical weathering then hydrolysis without necessarily involving C? Provide at least some references on this glacial weathering topic, which is the primary one related to this review.

211 DSi instead of dSi

Fig. 4c some comparison with non-glacial coastal environment could be useful to compare (as done e.g. fig 2a for ASi)

REVIEWERS' COMMENTS:

Reviewer #1 (Remarks to the Author):

We thank Reviewer #1 for their thorough summary and positive comments. We address their concerns below.

[...]Concerns:

1-This manuscript evokes a possible evolution of Si cycling in glaciated environments submitted to climate change. This is a crucial point. Unfortunately, the present version is too vague on this crucial point, as far as the potential impacts of changes in temperature and nutrients are concerned. They might be different for the arctic and antarctic environments.

We appreciate that we may have come across as vague in our discussion of this important matter. However, we are only just beginning to understand and quantify the impacts of climate change on this mechanism for nutrient supply to downstream ecosystems and need to be clear in our uncertainties at this point. To this end, we have expanded our final section, *How will Si cycling in glaciated environments change in a warming world?*, so that Line 592 now reads: "*However, there are several outstanding questions surrounding the mechanisms and influences of Si cycling within glaciers themselves that – at this stage – severely limit the confidence we can place on future predictions, including the role of supraglacial processes, or the connectivity, lithology and mineralogy of the subglacial environment. There are also ongoing compounding and multiple stressors acting on important biological and abiological processes across the land-ocean continuum, which are likely to differ between the Arctic and the Antarctic. For example, the temperature and nutrient*

content of Arctic seawater is likely to change, with continued atmospheric warming and the increase in marine input from the lower latitudes ("borealisation")...."

Note that we have emphasised the likely differences, as well as similarities, between the Arctic and Antarctic also on Line 568 which now reads: *"Furthermore, the impact of glacially-sourced DSi on productivity is likely going to differ between the Arctic and the Antarctic, given the latter generally experiences higher background seawater [DSi] in surface waters due to mixing with Si-rich Circumpolar Deep Water. However, in addition to large areas of the Arctic (e.g., Svalbard⁴⁰) and subarctic³⁹, there is evidence for at least seasonal silicon limitation in surface waters in part of the Southern Ocean⁷², and around some subantarctic islands (e.g., Scotia Sea⁷³), largely due to diatom uptake."*

2-That silicic acid delivered to coastal water through the Si glacial weathering can support a "substantial proportion of diatom growth" is mentioned in the abstract. Clearly, this is to be demonstrated, particularly for the Southern Ocean where much of the silicic acid input to surface waters comes from below, i.e, from upwelling of silicic acid rich deep water.

We agree that this source of silicic acid is likely going to be more important in the Arctic than the Antarctic and have now specified this in the abstract (indeed, we have demonstrated it to be the case off Southwest Greenland, as discussed in the manuscript). Also see comment above.

Reviewer #2 (Remarks to the Author):

Hendry and colleagues present a comprehensive review of Si behavior in glaciated margin systems with an intention of highlighting the insights related to benthic fluxes that can be gleaned from silicon isotopes specifically. As part of the review, the authors provide a synthetic model on which they perform a sort of sensitivity analysis that illustrates benthic Si fluxes along an offshore gradient.

Overall, I think this would be a useful paper for the community and I appreciate the idea of putting it all together. My main overarching comment is that I was not entirely clear on what NEW insights stem specifically from isotopes and what come from the sequential extraction based concentration measurements and what come from adding a model. If the goal is to highlight isotopes specifically, it may have missed. My impression was that it provides a more holistic picture of the near shore dynamics on glaciated margins with the measurements and models together.

Many thanks for the positive comments. We originally structured the manuscript around sections where we would introduce the background and challenges, before showing how stable and radioactive silicon isotopes could be useful as tools for addressing these challenges. We have restructured the abstract to include the following: *"Here, we review the biogeochemical complexities of glaciated margins and the insights into this system that silicon isotope geochemistry offer..."* to explain to the reader our intentions.

I also list editorial comments.

line 105-106... dissolution incubations recommended for light-dark cycles and T

differences... presumably these are meant to understand biological activity... but I wonder about the light component. In sediment, if they are buried shallow enough to receive light, only the thin layer of the uppermost sediment would be exposed? Is the hypothesis that this boundary layer hosts benthic diatoms that might accelerate A Si dissolution?

The experiments with light-dark cycles were not carried out by our research group but by Zhu et al., 2024. We assume that the idea was that the microbial assemblage might be sensitive to light conditions. Whilst the reviewer is correct that buried microbes would be entirely in the dark, it could be that water column and shallow surficial core-top sediment microbes could be exposed to light-dark cycles, even in low PAR conditions in turbid glacial river plumes.

Figure 2 caption..."See Supplementary Information for reactive silica (ASi) sequential extraction data: Maybe elaborate somewhere about this because I do not know why I am being directed to look for it.

This caption has now been changed to include: "*These plots show the total ASi extracted using a weak-alkaline leach, but additional information can be obtained about other reactive silica pools using sequential extractions (See Supplementary Notes...)*"

line 114: Box one discusses silicon isotope values or measurements, add either word after isotope.

Done.

Throughout the ms discussion of "light" isotopic signals would be more accurately portrayed as low values... as that is what is measured. I know this may seem pedantic but review articles are broadly read and adhering to strictly correct terminology is useful for linking this work to past and future work.

We have gone through the manuscript to ensure that "light" vs. "low" (and "heavy" vs. "high") are used appropriately i.e., "light" is used when referring to compositions or material, and "low" is used when referring to actual values.

line 116-117: Something seems off here: Glacial<non-glacial so parenthetical text is backwards?

Many thanks for spotting this – it is now corrected.

line 119-120: "appears to change over a melt season", how does it change (e.g. increased or decreased values with increased or decreased DSi?) and what does connectivity mean?

We have now clarified this sentence, which reads: "*Where data are available, the isotopic signal of DSi in glacial runoff appears to change over a melt season, trending towards lower $\delta^{30}\text{Si}_{\text{DSi}}$ values later in the season, reflecting the evolution of hydrological conditions and channel connectivity in the subglacial environment*"

Line 127: Signature should refer to an endmember: Replace light isotopic signature with low isotope values since $\delta^{30}\text{Si}$ of glacial silica, although low in value, varies.

We agree that “signature” should refer to an endmember. However, we argue that the glacial ASi composition is actually relatively invariant (Hatton et al., 2019), and does indeed represent an endmember input. As such, we have kept “signature” in this sentence.

Lines 135-139: While it is in the weeds, it seems like the size distribution matters in this case of crushing and re-crushing materials as well as Intense (longer) grinding mentioned on line 143 and for estimating something like reactive surface area either for modeling or experimental work.

Many thanks for this comment. The reviewer is correct that crushing and re-crushing has an impact on grain size. This is something we measured in our 2021 paper on the crushing experiments. However, we found no link between grain size and isotopic fractionation, which we attributed to: i) isotopic heterogeneity within the starting material, ii) natural pre-weathering of rock material, and/or iii) ASi formation and dissolution. We have added the following to line 176 to discuss this further: “*The mean grain size was smaller in the samples that were crushed for longer, but did not seem to result in greater fractionation, attributed to isotopic heterogeneity within – or pre-weathering of – the parent rock, and/or formation and dissolution of ASi.*”

line 150-151: I think this means $\delta^{30}\text{Si}$ values were linearly related to % glacial cover in the watershed?

This has been changed on line 186 for clarification to: “*the $<0.45\mu\text{m}$ fraction was characterised by isotopically light DSi that was linearly related to the percentage glacial cover in the watershed primarily due to a previously unidentified colloidal-nanoparticulate (0.02-0.45 μm) phase*”.

Lines 160-163: Confusing because it seems like the authors are on one hand referring to silicate weathering related CO_2 drawdown as something that every reader understands so well that it does not need to be discussed or even referenced but yet at the same time they parenthetically state it is not well defined. Do they mean the process, the magnitude, what?

This sentence has now been clarified on line 196 to read: “*Although there are inherent geochemical links between subglacial silicate weathering, oxidative weathering and atmospheric CO_2 uptake/release³⁷, we also need to consider downstream processes for a full picture of how glacially mobilised Si links with carbon cycling and climate*”.

Line 173, remove the parentheses.

Done.

Line 181-182, delete text in parentheses: measurement nuance and not needed in a discussion of what forms are in fjord sediments/source D_{Si}.

Done. However, we feel that this is still an important point to make, but we have moved this sentence to line 284, where it is more appropriately placed.

Line 193: I think it is more straightforward to say "why does D_{Si} appear to behave conservatively in fjords?" because given the above paragraph talking about D_{Si} as a major nutrient stimulating diatom growth, it just makes to sense to ask if it behaves conservatively.

We thank the reviewer for this suggestion, which we have acted upon.

Line 202: replace word location with "surrounding conditions" or something that describes more than just lat/long but things that may contribute to variations in nutrient supply, glacial proximity etc...

Done.

Line 221: "impacting the general inverse relationship", between what? given you have D_{Si} increasing and A_{Si} dissolving and Salinity varying. Be specific

We have clarified this sentence on line 277 to say: "*There is some field evidence for an increase in fjord D_{Si} concentrations with salinity (where salinity <10) (e.g.⁴⁴) rather than decrease, which could be explained by A_{Si} dissolution*".

Lines 236-240: Show predictions in Fig. 3 so reader can assess deviations from predicted.

Done. To help declutter this figure, we have cut out panel c, so now only show the isotope data vs. $\ln([D_{Si}])$ and vs. $1/[D_{Si}]$, which are more important in illustrating the key points in the text. See further comments in response to Reviewer #3.

Lines 265-267: What is the role of sediment accumulation rate on this relationship? the D_{Si} in the upper 50 cm of shelf has longer to accumulate or b_{Si} longer to dissolve.

This is a very good point, and one that we can address in the modelling, as mentioned (and now expanded upon) on line 527 onwards: "*We constructed a thought experiment to start illustrating the sensitivity of the earlier discussed diagenetic process interplay that controls benthic Si cycling in fjord systems to along-fjord gradients in bottom water conditions and sedimentation rates, which will be critical in determining the residence time of particles at the sediment-water interface*"

Figure 4 caption: "for porewater profiles and core incubations respectively. Unclear if respectively is supposed to mean with respect to the slash. maybe write as cores used for porewater profiles/core incubations ?

Many thanks for pointing out this ambiguity. This has been clarified as suggested.

Lines 398-403: I would recommend removing the mention of the Fe isotopes as they are not discussed again or here in detail and they do not add to the understanding that evidence exists for links between Si fluxes and dissolution/precipitation of Fe oxy hydroxides.

We respectfully disagree with the reviewer here. The Fe isotope inclusion is, by intention, brief as the focus is on Si, and it is meant to demonstrate the power of pairing isotope systems to understand Si dynamics in the benthic environment (reflected in the title of this section - *Combining isotope systems in early diagenesis*). We have added some text on line 497 to reemphasize this point, particularly pointing out that the lowest $\delta^{56}\text{Fe}$ (implying Fe microbial reduction) intersects with the highest $\delta^{30}\text{Si}$, implying Si adsorption onto Fe oxyhydroxides. We think it is important this is highlighted to help stimulate future research into this Si-Fe coupling.

Line 434/Line 452. Models are thought experiments. No need to add language that makes it less clear what you did. Sensitivity analysis might be more useful than Gedankenexperiment.

We respectfully disagree with this statement: models and thought experiments (or Gedankenexperimente) are different concepts. By definition, a thought experiment is a mental exercise to analyze hypothetical scenarios, while a model is a mathematical (or physical or conceptual) representation used to simulate and study real-world systems.

We would argue here that the model functions as a thought experiment, used to conceptually test ideas to explore conceptual patterns of Si cycling along the land-ocean gradient.

A sensitivity analysis, in contrast, is a method used to determine how changes in variables (background conditions, forcings, parameters) affect the specific output of a model. This typically entails a systematic (i.e., large ensemble over carefully defined parameter space) and statistical analysis. As such, we would argue that the term "thought experiment" is the better fit here and have decided not to make the suggested change.

Line 442: "reasonable" implies a value judgement, why not range of bottom water conditions.

This has been corrected as suggested.

Table 1: include something that characterizes the environment of cores A, B, C in table, e.g. fjord, shelf, slope.

Many thanks for this suggestion. Done.

Line 502: Methods to address questions: Can you combine this section with the section above and make a table of methods/tools. Reading lists as text is difficult at any point but at the end of a paper it is nearly impossible. A table would be more

valuable to readers.

Many thanks for this suggestion. We have decided to keep this section as a narrative rather than a table, as we already have Box to explain silicon isotope methods. However, we would be happy to discuss this with the editorial team if a table is preferred.

Reviewer #3 (Remarks to the Author):

This paper presents an overview of recent advances on subglacial processes in the biogeochemical cycling of silicon. Many studies have been published in the last decade and it is indeed a 'hot' topic of interest to several scientific communities (climate / ocean / biogeochemistry / geosciences) because (i) it is relatively new (weathering in cold climates has often been overlooked because it was thought to be minor compared to humid and warm climates) and, (ii) also because of the dramatic changes these polar environments are currently facing due to global warming. So, this paper is welcome, and since most of the recent work has been published by this group in (too?) many different papers, the review really helps to get a much clearer overall picture of the processes. The paper is very interesting and is a nice and useful synthesis of these works.

Many thanks for these positive comments – they're very much appreciated!

However, I have some comments below to make the synthesis clearer and more appropriate. I can summarise my main concerns here: (i) homogenisation of nomenclature (e.g. silica vs. silicon; Si isotope notation; better separation/definition of processes and areas along the glacier-ocean continuum fjord/shelf/mixing....); (ii) more appropriate use of Si isotope models (by definition data from different Si isotope systems cannot fit the same isotopic trend); (iii) improve the synthetic Fig.5 so that it is more informative and it better reflects both the complexity of the processes discussed and the consequences for the silicon (isotope) cycle. In this respect, some processes are not sufficiently discussed (e.g. reverse weathering and land processes); (iv) the methods on sediment leaching as well as the modelling part would need some additional information in the supplementary material.

Many thanks for these constructive comments, which we will address below.

Nomenclature and notation

- Silica vs silicon. Silica = SiO_2 . Silicon = Si. This looks trivial but is too often incorrectly referred to, including in this paper. In the dissolved form, Si is under silicic acid and can be referred to as DSi. Sometimes this DSi comes from the dissolution of silica (biogenic, amorphous or even from quartz), sometimes not and comes from aluminosilicates (amorphous or crystalline) or from desorption, i.e. not from silica so the term dissolved silica is incorrect. Please do remind this (e.g. in the box) and try to respect it in this review (e.g. lines 82 and 146 silica should be replaced by silicon).
- Si isotopes (lines 330, 426, 546, box etc.). Avoid the notation d^{30}DSi or d^{30}ASi which is chemically confusing (superscript 30 associated with D or A and not Si is

incorrect), and rather keep the notation $d_{30Si}DSi$ and $d_{30Si}ASi$ (with DSi and ASi as subscripts) as this is the main notation used in this ms.

We have been through the manuscript to ensure that all usages of the terms are appropriate. Apologies for those that slipped through 'the net' in our original submission.

Different processes along the glacier - to - ocean continuum

- Lines 167 before the subsection on fjord. There is no mention on what can happen for land terminated glacier to the glacial material before it reaches the fjord / coast via the proglacial river. The presence of lakes for instance could dramatically change the particulate content downstream (settling); it might also change the DSi concentration and Si isotopic composition in case of freshwater diatoms Si uptake. Even without lake, glacial plains can be wide and under low slope forming meanders favoring settling in the proglacial rivers running at low speed. Has this aspect been studied in the different areas compiled here? Could it explain variability within land terminated glaciers as well as in between the 2 categories summarized in Fig. 5 (land-terminated or sea-terminated)? This is a well-known key property controlling the supply of particulate to the coast in non-glacial environments, why not for glacial environments?

This is an excellent point, regarding an understudied environment. To address this, we have included the following on line 220: *“Also, in the case of land-terminating glacial fjord, any glacially-sourced nutrients will also be exposed to additional processing in the pro-glacial watershed system, which could include glacial lakes in addition to rivers³⁸. A proportion of both DSi and ASi will likely be trapped, at least in part, within the fjord (and/or pro-glacial) environment before reaching – and so potentially supporting – coastal ecosystems.”*

- Section fjord vs. continental margin sediments. Similarly, it's unclear where is the separation all over the manuscript between fjord and continental margin. Is it the geomorphology as it appears since no other precision is given? However, what does matter is the mixing zone between SW and freshwater. Depending on the site, I assume this zone might be within the 'fjord' zone or as plume over the continental margin/shelf; of course, this also depends on type of glacier: land-terminated or sea-terminated. From the synthetic Fig. 5, the category is based only on this type of glacier: fjord is displayed as the interface between land and shelf but this part of the scheme does not look like a fjord and looks more like a plume extending into the sea. I would recommend to make a separation as 3 zones 100% freshwater, 100% seawater and the mixing zone in between; or clarify the choice / illustration / sections made here. This confusion extends to the third category 'coastal system and beyond' from line 310. The fig. 1, fig. 5 and the text should be organized accordingly for a better understanding of the authors' purpose.

Many thanks for this comment, and we would be happy to clarify our aims. We very much intended to emphasise that the system is a continuum, and that there is no clear geographical demarcation between the different coastal “zones” in terms of seawater properties. In the text, we have clarified the separation between fjord and continental shelf on line 214 onwards, which now reads: *“In most glaciated environments, the DSi and ASi released in glacial runoff will flow in the coastal*

ocean via fjords, which are typically separated from the continental shelf by a shallower sill. Herein lies a key issue: fjords are not only conduits but also active and highly heterogeneous biogeochemical reactors, with multiple input sources, including non-glacial rivers and the ocean itself e.g., see ³⁸, and abiotic and biotic processes occurring across strong but temporally and spatially variable salinity gradients that form a continuum with the open ocean¹³⁹.

We have also clarified our subheadings to be consistent in our terminology.

In the summary figure (Fig. 5), we have labelled the different zones (ocean, shelf, fjord, land), for illustrative purposes (we have made the font size bigger here, to make this clearer to the reader). We decided not to include any indicative salinities in this figure, as this can be highly variable laterally and with depth (as the reviewer remarks upon). This variability is now noted on line 219 (see above). It should also be noted that for most of the fjord system, the bottom waters (i.e., at the sediment-water interface) will be saline, which is now clarified in the supplementary information (also see comment below).

For Fig. 1, we have clarified the marker symbols in the figure caption (the symbols differentiate terrestrial i.e., 100% freshwater, from “*fjord and marine*”).

Regarding figure 5, we have two “end-member” glacial fjord types represented in the schematic (land-terminating and marine-terminating). We believe marking a salinity gradient on the plot would not be beneficial, as the geographical location of mixing zones can change dramatically between seasons and between years. Please see further comments on this figure below.

- 280 and model. I appreciate the discussion on the role of Fe-Mn oxide to adsorb significantly Si, but I miss a section on 'traditional' reverse weathering, i.e. formation of authigenic secondary minerals (sensu Michalopoulos & Aller, 2004) which was proposed on coastal sediments. It has been suggested to (partly) control also $\delta^{30}\text{Si}$ in open polar ocean sediment at low $T^\circ\text{C}$ (Closset et al., 2022). While this is briefly mentioned here, I see no reason why to favour the Fe-Mn and Si adsorption process compared to reverse weathering. In fact, more discussion appears at the end of the ms (from line 408) in the section on the Biogeochemical Reaction Network Simulator where it includes authigenic minerals formation and implicit representation of adsorption/desorption. This is confusing. I suggest to put the discussion on BRNS earlier when these processes are tackled.

Many thanks for this comment. We have attempted to balance out the discussion on uptake processes by adding in more regarding reverse weathering, which we do – indeed – find evidence for in glacial sediments. It might well be that there is a linkage between the two processes, with adsorption favouring the pre-concentration of Si on mineral surfaces to promote in situ precipitation, as suggested in the manuscript on line 359.

Line 347 now reads: “*One possible mechanism is that authigenic precipitation of secondary minerals take up DSi in addition to metal cations, often termed reverse weathering⁴⁹. An alternative mechanism relates to the supply of reactive metal minerals – largely iron (Fe) and manganese (Mn) oxyhydroxides – from subglacial*

weathering... New sediment sequential extraction methods, combined with stable isotope and elemental analyses, are beginning to shed light on different operational pools of reactive silica in marine sediments, including a likely component of poorly-crystalline authigenic silica and adsorbed Si in the weak acid-extractable fractions (see Supplementary Information)".

- Fig. 5 is a pretty scheme, but is too general. It summarises too basic and trivial information, much simpler than the previous studies which are nicely discussed in this synthesis ms. I would suggest to complexify it to better reflect all these works by e.g. making different panels for different types of glacial environments (mixing zone, upstream fjord...) and/or adding $\delta^{30}\text{Si}$ variability and/or adding Si flux or Si contribution ranges from different processes as estimated from the model (e.g. by pie chart).

We have taken the comments of the reviewer on board, and tried to add some additional data to the schematic. To do so, we have added sketches of the typical relative changes in DSi concentration downcore for each of the cores A-C. We refer the readers to the model results in the new figure caption. We feel that there is a balance to maintain with such figures: we need to summarise key points without having too much information, and we hope that we have struck that balance here.

Si isotopes models

- Fig.3 is not adequate to support the discussion and is at least partly misleading. In 3a and 3b, as reported, it is true that we expect a linear relationship for mixing (3a) or for diatom uptake in a closed system (3b). However, such a relationship would only be valid for a single isotopic system (as simply imposed by the isotopic mixing or Rayleigh equations), i.e. we should have linearity for each site, not only for each type of symbol, but also within a symbol such as glacial rivers or fjords where there are different sites. I acknowledge that glacial ASi $\delta^{30}\text{Si}$ could be considered the same everywhere (btw a key finding and very nice contribution of all these previous studies and well highlighted here), but DSi concentrations and $\delta^{30}\text{Si}$ -DSi of end members (for 3a) or DSi sources (for 3b) are highly variable (as are ASi concentrations). The scale makes these panels very busy, and they are not evidence of what is claimed, since potential linearity is not discussed per site, only rejected for the whole data, which is trivial and not relevant.

This is a very valid point. However, we have been careful to select data from one geographic location i.e., SW Greenland. The fjord data are all from one fjord system, and the coastal data are all from adjacent ocean sampling (with Fram Strait data included as an "upstream" comparison). We have now edited the figure to only include glacial meltwater data from SW Greenland as well. This has now been clarified in the figure caption (and we have checked key location names are now plotted on the map in Figure 1). We completely agree with the reviewer that it could have been misleading without this clarification, especially as we emphasise the geographic variation in our measurements in other parts of the manuscript, e.g., Figure 1).

Similarly, for 3c, each system should have its own mixing curve which is not shown. For the revised paper, one possibility could be to make 3 small panels for each site and put them in the supp mat, although I find the supp mat already extremely long....

The other possibility might be to just delete this figure without changing much of the text and just refer to the original articles.

We have included mixing curves as requested by Reviewer #2 into the new Fig. 3, which we believe is valid as all the samples are from the same geographic location.

- Fig. 4d See comment above on Fig. 3 there is no reason why all this data would plot on the same Rayleigh system; even within the same symbol type (e.g. just for Southern Ocean sites, the D_{Si} concentration varies by a factor of 2 and d_{30Si} by almost 1 ‰ depending on the cores and bottom water D_{Si} are order of magnitude higher than the ones in the Arctic, so the PW isotopic data simply cannot plot on the same Rayleigh). Nothing can be expected from such plot with data from so different origins. Note that I do agree with the claim that heavier d_{30Si} of PW in the proximal sites suggest preferential immobilization of light isotopes but again, this conclusion does not come from Fig. 4d in its current form.

This is a valid point. We included the different locations in Fig. 4d originally to show the spread in the observations, but we have removed this panel of the figure now to avoid misleading any readers inadvertently. As the reviewer suggests, the key point can be shown from panels a and b, which are collected from the same geographic location.

Methodology in supplementary material

S1 text and table: Methodology is not the main topic of this paper, which I somewhat regret as there is no agreed method for measuring Si isotopes from coastal sediments, especially when high non-biogenic amorphous silicon is present. It should at least be highlighted here that several leaching/sieving methods are used and briefly discussed whether this could explain some of the variability found in different studies, e.g. in a table summarising these methods and then discussed. For example, why is Iceland so variable and higher (lithology is mentioned but this is not convincing)?

We respectfully disagree that lithology is not a convincing argument for the differences we observe. We acknowledge that there are different methodologies used, with subtle differences, that could explain some of the variability (and we completely agree that more work needs to be done to understand the extraction processes in more depth – we have commented on this now in the text). However, all of the analyses shown in the plot were carried out in my group using the same methods (now clarified). The few groups that have done extractions from similar locations agree well with our findings. This has been included in the supplementary information:

“Whilst there are differences in the methodologies between laboratories, in the situations where there are two different groups that have studied glacial sediments from a similar location, bulk ASi data are in broadly good agreement e.g., Svalbard e.g., this study and ⁸.”

Model

This part is quite dense, and only described in the supp material. The idea is relevant and interesting and have been (partly?) published in a separated article. Since (i) it is

not the core of this manuscript (ii) it is only shortly presented in the supp mat (iii) it's not my domain of expertise, I cannot fully evaluate this part. I do have some comments though:

- Clarify what is new in this modelling part compared to Ward et al. 2022 (btw it's erroneously cited as Ward et al. 2020, while it's the 2022 in the reference list; please check) et Ng et al. 2022?

Many thanks for spotting the error with the Ward et al., citation. This has been fixed. The main/key difference is in the parametrization of FeSi reactions. 1) In both previous papers, that was somewhat data-driven and implemented quite implicitly as a desorption only (Ward et al.) and adsorption/desorption (Ng et al.) factor. 2) Here, the reactions are more explicit. They account for a rate constant each that can be treated as constant or have an exponential decay; and a fractionation factor that can be changed to better explore how FeSi dynamics impacts DSi- $\delta^{30}\text{Si}$; and it links to DSi production/consumption. It also has a better control of the depth zonation of where adsorption and desorption occur.

The caveat that still remains is that it treats these processes assuming a kinetic formulation and relies on knowing *a priori* the iron redox zonation. So that's the next jump!

This has now been clarified in the Supplementary Information.

- I wonder if there are any data for the 4th assumption of the model (supp mat: "Iron-adsorbed silicon (FeSi) occurs as a pre-formed silicon phase in the water column."). Is it an important feature of model output? Please provide some reference / justification.

This parameter is not critical for the model output (ASi and BSi are the most critical), but we are simply describing how it diminishes along the fjord head-opening transect.

- With reference to my previous comments on the different categories chosen, how are the compositions of these cores affected by the location of the mixing between SW and freshwater? Is core B implicitly under the mixing zone?

The main difference in composition between the cores is the input (and how it changes) of BSi and ASi. We assume a gradient along the fjord, where ASi decreases from the fjord head out and BSi does the opposite. Core B is not in a mixing zone (indeed, it should be noted that all sediment cores experience saline conditions at the sediment-water interface, given freshwater will be found higher up in the water column). This is explained in the tables in the Supplementary Information.

- Would it be possible to plot real data on fig. S2 to validate the model output??

We do not feel it is appropriate to include 'real' data in these plots, for the reason that these are conceptual thought experiments (see comment above to Reviewer #2). We are not attempting to reproduce specific sites/data but to illustrate the conceptual pattern. Please note that the reader can refer to previous fjord/polar sediment model studies we have carried out and that all include model-data fits, which highlight the

need to have the model structure the way it has been designed. As such, we have decided, with respect, not to act on this suggestion.

Minor comments

94 Fig S2 is cited instead of Fig. S1.

We were referring to the map in Fig. 1, which has been clarified.

160-162, needs to be developed since the link is not trivial. Is CO₂/H₂CO₃ involved in this weathering? Could it be only mechanical weathering then hydrolysis without necessarily involving C? Provide at least some references on this glacial weathering topic, which is the primary one related to this review.

This has now been addressed (see comment to reviewer 2 above), including a reference as requested.

211 DSi instead of dSi

Corrected.

Fig. 4c some comparison with non-glacial coastal environment could be useful to compare (as done e.g. fig 2a for ASi)

Done. See new Fig. 4d.

References:

Hatton, J. E., Hendry, K. R., Hawkings, J. R., Wadham, J. L., Opfergelt, S., Kohler, T. J., ... & Žárský, J. D. (2019). Silicon isotopes in Arctic and sub-Arctic glacial meltwaters: the role of subglacial weathering in the silicon cycle. *Proceedings of the Royal Society a*, 475(2228), 20190098.

Zhu, X., Hopwood, M. J., Laufer-Meiser, K., & Achterberg, E. P. (2024). Incubation experiments characterize turbid glacier plumes as a major source of Mn and Co, and a minor source of Fe and Si, to seawater. *Global Biogeochemical Cycles*, 38(10), e2024GB008144